# PRDM1/BLIMP1 induces cancer immune evasion by modulating the USP22-SPI1-PD-L1 axis in hepatocellular carcinoma cells

Qing Li[1,10], Liren Zhang[1,10], Wenhua You[2,3,10], Jiali Xu[4,10], Jingjing Dai[5,10], Dongxu Hua[6,10], Ruizhi Zhang[1], Feifan Yao[1], Suiqing Zhou[1], Wei Huang[7], Yongjiu Dai[1], Yu Zhang[7], Tasiken Baheti[7], Xiaofeng Qian[1], Liyong Pu[1], Jing Xu ●[8] ✉, Yongxiang Xia ●[1] ✉, Chuanyong Zhang ●[1] ✉, Jinhai Tang ●[9] ✉ & Xuehao Wang ●[1] ✉

Programmed death receptor-1 (PD-1) blockade have achieved some efficacy but only in a fraction of patients with hepatocellular carcinoma (HCC). Programmed cell death 1 ligand 1 (PD-L1) binds to its receptor PD1 on T cells to dampen antigen-tumor immune responses. However, the mechanisms underlying PD-L1 regulation are not fully elucidated. Herein, we identify that tumoral *Prdm1* overexpression inhibits cell growth in immune-deficient mouse models. Further, tumoral *Prdm1* overexpression upregulates PD-L1 levels, dampening anti-tumor immunity in vivo, and neutralizes the anti-tumor efficacy of *Prdm1* overexpression in immune-competent mouse models. Mechanistically, *PRDM1* enhances *USP22* transcription, thus reducing *SPI1* protein degradation through deubiquitination, which enhances *PD-L1* transcription. Functionally, PD-1 mAb treatment reinforces the efficacy of *Prdm1*-overexpressing HCC immune-competent mouse models. Collectively, we demonstrate that the PRDM1-USP22-SPI1 axis regulates PD-L1 levels, resulting in infiltrated CD8⁺ T cell exhaustion. Furthermore, *PRDM1* overexpression combined with PD-(L)1 mAb treatment provides a therapeutic strategy for HCC treatment.

Recently, immunotherapies have achieved tremendous clinical breakthroughs in several cancers. Programmed death ligand 1 (PD-L1, namely CD274) is widely expressed in various malignancies and induces immune evasion by targeting its ligand, programmed cell death 1 (PD-1) on activated T cells, resulting in T cell exhaustion[1–4]. Blocking

PD-1/PD-L1 signaling significantly improves T cell responses and achieves a striking clinical response in various advanced malignancies. However, most patients, especially patients with HCC, are unresponsive to the blockade of the PD-1/PD-L1 axis[5,6]. Thus, extensive efforts are needed to identify reliable predictive biomarkers. Tumoral PD-L1

[1]Hepatobiliary Center, The First Affiliated Hospital of Nanjing Medical University, Key Laboratory of Liver Transplantation, Chinese Academy of Medical Sciences, NHC Key Laboratory of Living Donor Liver Transplantation (Nanjing Medical University), Nanjing, Jiangsu Province, China. [2]School of Chemistry and Chemical Engineering, Southeast University, Nanjing, Jiangsu Province, China. [3]Department of Immunology, Key Laboratory of Immune Microenvironment and Disease, Nanjing Medical University, Nanjing, Jiangsu Province, China. [4]Department of Anesthesiology and Perioperative Medicine, The First Affiliated Hospital of Nanjing Medical University, Nanjing, Jiangsu Province, China. [5]Department of Infectious Diseases, The First Affiliated Hospital of Nanjing Medical University, Nanjing, Jiangsu Province, China. [6]The First School of Clinical Medicine, Nanjing Medical University, Nanjing, China. [7]Department of General Surgery, The Friendship Hospital of Ili Kazakh Autonomous Prefecture, Ili & Jiangsu Joint Institute of Health, Ili, China. [8]Department of Oncology, The First Affiliated Hospital of Nanjing Medical University, Nanjing, Jiangsu Province, China. [9]Department of General Surgery, The First Affiliated Hospital of Nanjing Medical University, Nanjing, Jiangsu Province, China. [10]These authors contributed equally: Qing Li, Liren Zhang, Wenhua You, Jiali Xu, Jingjing Dai, Dongxu Hua. ✉e-mail: xujing7901@jsph.org.cn; yx_xia@njmu.edu.cn; zcy13951673178@163.com; jhtang@njmu.edu.cn; wangxh@njmu.edu.cn

expression is considered an underlying biomarker of clinical response; however, the mechanisms underlying the regulation of constitutive PD-L1 expression are complex and need to be clarified further. In addition, tumoral PD-L1 expression alone is not a reliable predictive biomarker of immunotherapy in most patients with HCC[5]. Therefore, uncovering the mechanism of PD-L1 regulation and developing additional predictive and therapeutic markers for PD-1/PD-L1-based therapies in HCC are necessary.

We and others have recently demonstrated that the PRDI-BF1 and RIZ homology domain (PRDM) family mediates a series of pathological conditions, especially in cancers. *PRDM1* gene (encoding for BLIMP1 transcription factor) is involved in diffuse large B cell lymphoma[7,8]. *PRDM2* is a tumor-promoting factor in multiple malignancies[9,10]. *PRDM3* impairs pancreatic tumorigenesis by regulating inflammatory responses[11]. *PRDM4* inhibits tumorigenesis in cervical carcinoma and also contributes to YAP-induced tumorigenesis[12,13]. Downregulation of *PRDM5* has been reported in multiple human cancers[14]. Our previous study also emphasized that *PRDM8* triggers antitumor effects in HCC[15]. Rare allelic forms of *PRDM9* can drive childhood leukemogenesis[16]. *PRDM11* silencing supports MYC-driven lymphomagenesis[17]. *PRDM13* overexpression inhibits glioma cell proliferation and invasion[18]. *PRDM14* promotes malignant phenotypes in various cancers[19-21]. *PRDM15* rewires metabolic pathways critical for sustaining B cell lymphomagenesis[22]. *PRDM16* acts as a suppressor in various tumors, including kidney and lung adenocarcinoma[23,24]. However, the role of the PRDM family in regulating anti-tumor immunity and immune molecules in HCC cells remains largely unknown.

Here, we show the role of *PRDM1*/BLIMP1 in the regulation of immune molecules expressed by HCC cells with respect to cancer immune evasion. We demonstrate that the PRDM1-USP22-SPI1 axis regulates PD-L1 levels, resulting in infiltrated CD8[+] T cell exhaustion. Furthermore, *PRDM1* overexpression combined with PD-(L)1 mAb treatment provides a therapeutic strategy for HCC treatment.

## Results

### PRDM1/BLIMP1 is a prominent regulator of PD-L1 in HCC

To screen the regulators of *PD-L1* expression among the PRDM family, correlation analysis using the GEPIA database revealed the key regulators of *PD-L1* expression. *PRDM1, PRDM2, PRDM3, PRDM4, PRDM5, PRDM6, PRDM8, PRDM10, PRDM11*, and *PRDM13* were identified as *PD-L1* expression regulators. Among these, *PRDM1* showed the highest correlation with *PD-L1* expression (Supplementary Fig. 1a). Therefore, we focused on the anti-tumor immunity of *PRDM1* through *PD-L1* regulation. To validate *PRDM1*-induced tumoral *PD-L1* upregulation in HCC cell lines, *PRDM1* was overexpressed using the LV-*PRDM1* vector in Hep3B cells and was suppressed in Huh7 cells using CRISPR Cas9-targeted mutation (sg*PRDM1*) (Supplementary Fig. 2a). PD-L1 upregulation at both mRNA and total and cell surface protein levels after *PRDM1* overexpression was confirmed in *PRDM1*-overexpressing Hep3B cells, with or without IFN-γ stimulation, and the opposite effect was observed in *PRDM1*-knockout Huh7 cells (Fig. 1a–e). To simulate the tumor microenvironment affected by *PRDM1* alterations, a 3D culture system was constructed using an IFN-γ pre-treated HCC cell line (Hep3B and Huh7 cells, 500 IU/ml) and pre-activated T cells (Fig. 1f). *PRDM1* upregulation impaired the CD8[+] T cell activation (CD8[+]GZMB[+] T cells) and T cell-mediated tumor cell killing activity (CD8[+]TNFα[+] T cells) in the co-culture, whereas *PRDM1* knockout had contrasting effects (Fig. 1g, h). The binding of IFN-γ pre-treated HCC cells (500 IU/ml) with PD1/Fc protein (green fluorescence) indicated that the PD-L1-PD1 interaction steadily increased over 12 h. With *PRDM1*-induced increased PD-L1 expression, *PRDM1* overexpression increased PD1 binding to HCC cells. *PRDM1* knockout-mediated decreased PD-L1 expression also diminished PD1 binding to HCC

cells (Fig. 1i, j). As T cell exhaustion due to PD-L1/PD1 inhibitory receptor signaling impairs T cell proliferation and effector functions, the relationship between tumoral BLIMP1 protein expression and exhausted T cells in HCC specimens was examined by profiling the content and function of CD8[+] T cells in tumor-infiltrating T lymphocytes (TILs) using flow cytometry in cohort 1 (40 patients with HCC showing diverse tumoral BLIMP1 protein expression). BLIMP1 protein expression was inversely associated with the proportion of CD8[+] T cells ($R = -0.4877$, $P = 0.0014$) and the activity (GZMB[+]) of CD8[+] T cells ($R = -0.4882$, $P = 0.0014$) in infiltrating CD45[+] cells. BLIMP1 protein expression was positively correlated with the proportion of exhausted CD8[+] T cells (PD1[+]) ($R = 0.3722$, $P = 0.0181$) in CD8[+] TILs (Fig. 1k, l). GEPIA datasets further validated that T cell exhaustion markers were positively correlated with *PRDM1* levels in HCC, indicating the TILs regulation of *PRDM1* expression (Supplementary Fig. 1b).

### Prdm1 promotes immune evasion by upregulating PD-L1 expression in immune-competent mice and HCC cells

Hepa1-6 and H22 transfected with *Prdm1* overexpression vector and knockout vector, respectively (Supplementary Fig. 2a), were inoculated into immune-competent mice (C57BL/6 mice). We expected that *Prdm1* overexpression in immune-competent mice would contribute to tumor proliferation and decrease overall survival (OS), and that *Prdm1* knockout would display slower tumor proliferation and better OS. However, no obvious differences were observed between the *Prdm1* or sg*Prdm1* groups and their corresponding controls in terms of tumor size and OS. Tumors from the subcutaneous HCC models were then transplanted into the livers of C57BL/6 mice to establish immunocompetent orthotopic models. The *Prdm1* and sg*Prdm1* groups and their corresponding controls showed comparable tumor sizes (Fig. 2a–f). However, flow cytometry revealed reduction of infiltrated CD8[+] T cells and decreased activity (GZMB[+]) of infiltrated CD8[+] T cells in the *Prdm1* groups in the subcutaneous transplanted model. Further, infiltrated CD8[+] T cell exhaustion assessed by PD-1[+] staining was significantly increased in *Prdm1* groups. Infiltrated CD8[+] T cells and the activity (GZMB[+]) of infiltrated CD8[+] T cells was obviously increased, and infiltrated CD8[+] T cell exhaustion was significantly reduced in the sg*Prdm1* groups (Fig. 2g–i). To examine other molecular mechanisms, we inoculated murine liver cancer cells (Hepa1-6 and H22) into immunodeficient mice (BALB/c nude mice); tumor proliferation was distinctly slower in the *Prdm1* group and was faster in the sg*Prdm1* groups than in the corresponding controls. Further, prolonged and shortened OS were observed in the *Prdm1* and sg*Prdm1* groups, respectively. Immunodeficient orthotopic models also confirmed that *Prdm1* overexpression resulted in smaller tumors, whereas the opposite effects were observed in the sg*Prdm1* groups (Fig. 2j–o). Moreover, no obvious differences were observed between the *Prdm1* or sg*Prdm1* groups and their corresponding controls in terms of Ki67 staining in immunocompetent mice. Nevertheless, *Prdm1* overexpression reduced cell proliferation, while *Prdm1* knockout promoted cell proliferation in immunodeficient mice (Supplementary Fig. 2b, c). Thus, *Prdm1* fails to impede tumor proliferation in immunocompetent mice but impairs tumor progression in immunodeficient mice. CCK8 assays verified that *PRDM1* suppressed proliferation in both human and murine HCC cell lines (Supplementary Fig. 2d). In T cell-mediated tumor cell-killing assays, *PRDM1* overexpressing HCC cells showed increased resistance to activated primary CD45[+]CD3[+] T cells isolated from PBMCs in vitro (Fig. 2p, q). Thus, T cell-mediated immune evasion impedes the anti-tumor effect of *PRDM1* overexpression.

To validate if *Prdm1* contributes to immune evasion through PD-L1 in vivo, tumors resected from C57BL/6 mice were subjected to IF and qRT-PCR. Significant upregulation of PD-L1 protein and mRNA expression was found in tumors from *Prdm1* groups, related to an

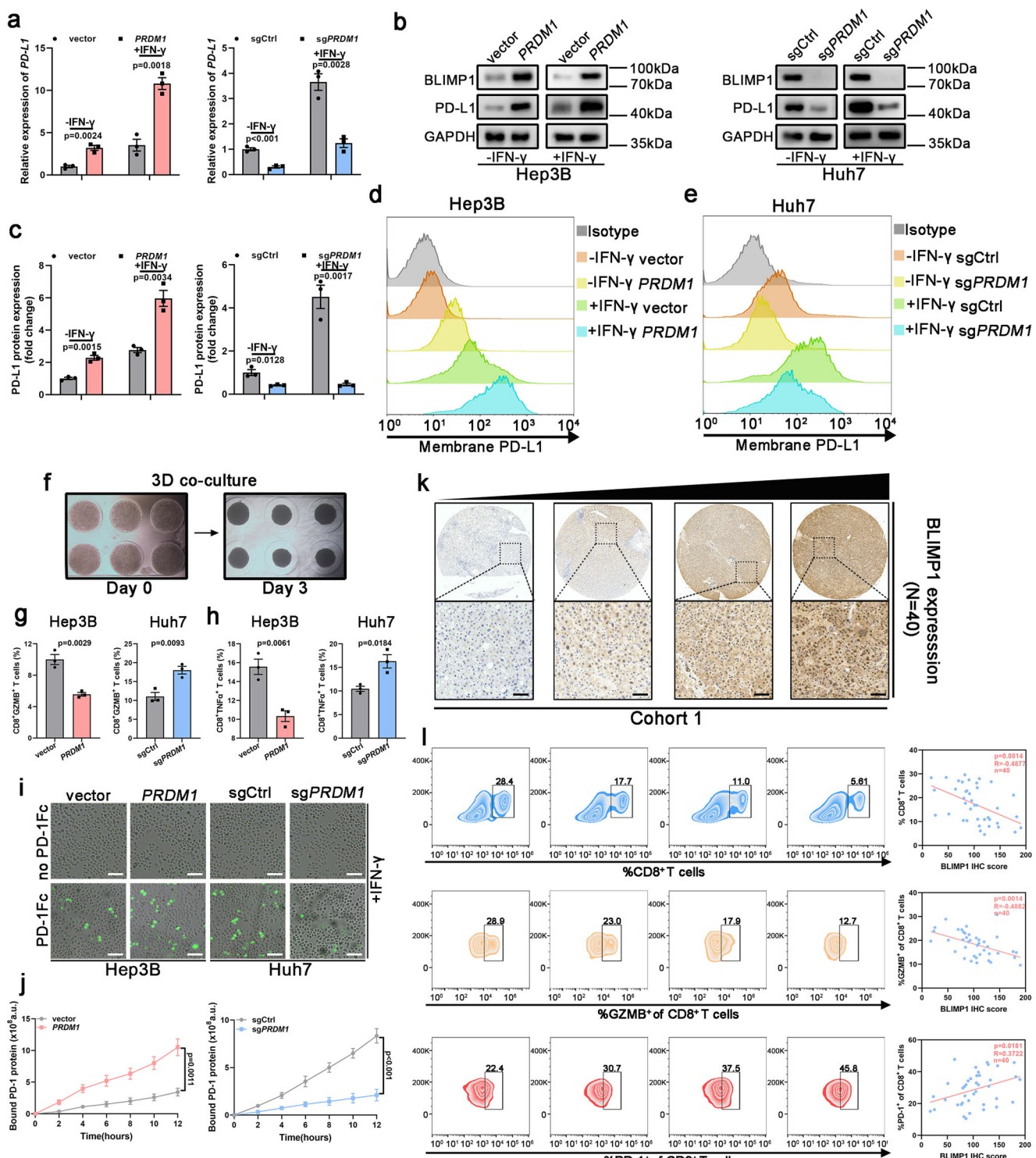

**Fig. 1 | PRDM1 upregulates PD-L1 expression in HCC. a** qRT-PCR of *PD-L1* expression in Hep3B cells or Huh7 cells with or without IFN-γ treatment. Data presented as mean ± SEM (*n* = 3 independent biological replicates). **b, c** Western blotting of PD-L1 expression in Hep3B cells or Huh7 cells with or without IFN-γ treatment. Representative of *n* = 3 independent biological replicates. **d, e** Flow cytometry analysis of PD-L1 expression in Hep3B cells (**d**) or Huh7 cells (**e**) with or without IFN-γ treatment. *n* = 3 independent biological replicates. **f** 3D-co-culture model using pre-activated T cells and Hep3B or Huh7 cells. **g** Flow cytometry analysis of CD8⁺GZMB⁺ cell content in pre-activated T cells following 72 h of co-culture with Hep3B or Huh7 cells. Data presented as mean ± SEM (*n* = 3 independent biological replicates). **h** Flow cytometry analysis of CD8⁺TNFα⁺ cell content in pre-activated T cells following 72 hours of co-culture with Hep3B or Huh7 cells. Data presented as mean ± SEM (*n* = 3 independent biological replicates). **i** Time-

lapse microscopic images revealing the binding of Hep3B or Huh7 cells with PD1 at 12 h. Scale bars, 100 μm. **j** Analysis of PD1/Fc protein binding on Hep3B and Huh7 cells at 2 h. Data presented as mean ± SEM (*n* = 3 independent biological replicates). **k** Immunohistochemical staining of BLIMP1 protein in HCC samples (Cohort 1). Scale bars, 100 μm. **l** Correlation analysis of tumoral BLIMP1 protein expression and tumor-infiltrating T cell content. Pearson analysis revealed a negative correlation between tumoral BLIMP1 protein expression and CD8⁺ T cells and the activity (GZMB⁺) of infiltrated CD8⁺ T cells in infiltrating CD45⁺ cells and a positive correlation between tumor BLIMP1 protein expression and the percentages of PD1⁺ cells in CD8⁺ TILs. *n* = 40 patients. P value was determined by unpaired two-sided Student's *t* test (**a, c, g, h, j**) and Pearson correlation analysis (**l**). Source data are provided as a Source data file.

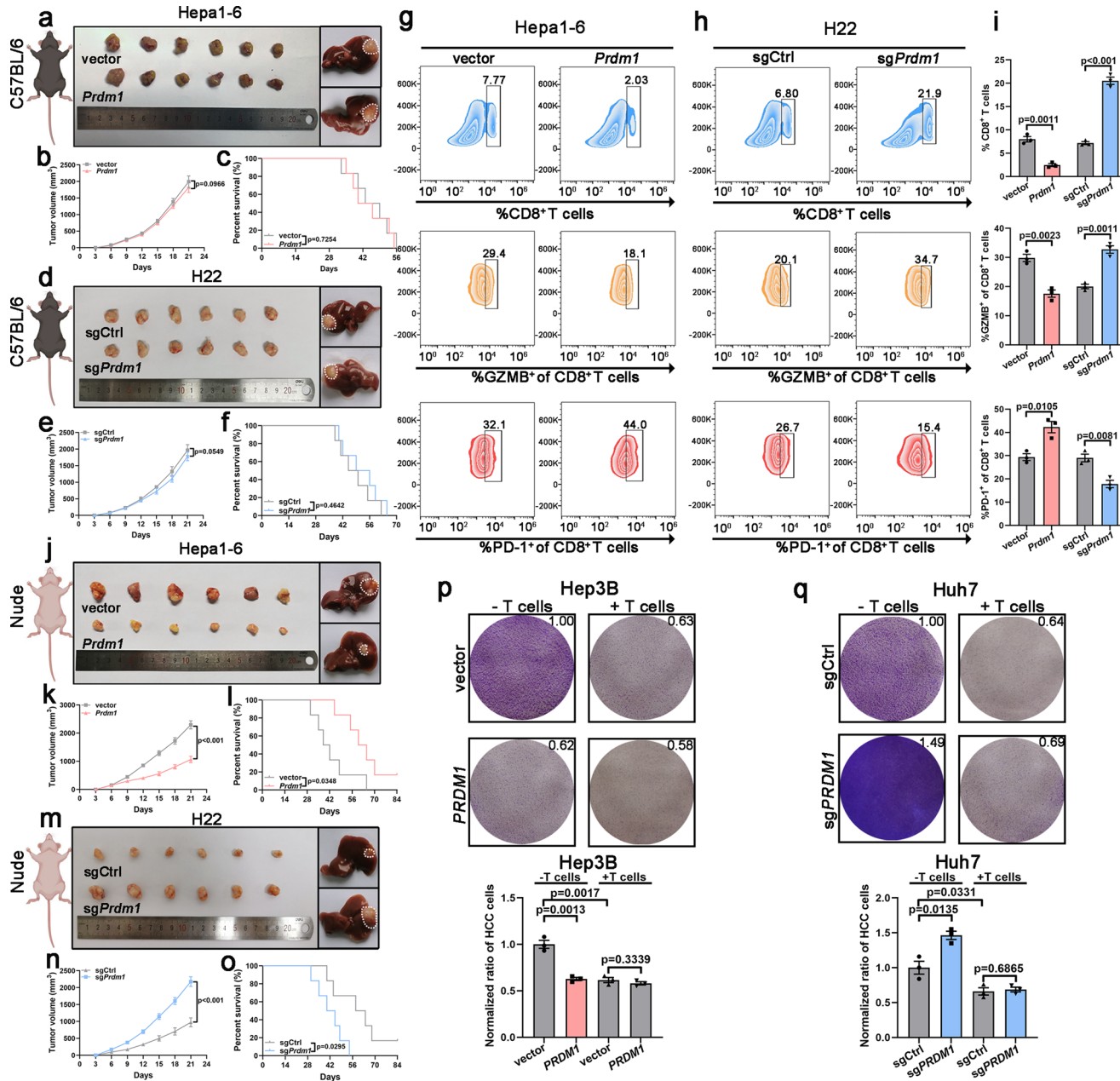

**Fig. 2 | Prdm1 attenuates its effects on tumor growth inhibition through PD-L1-induced tumor immune evasion in immune-competent mice. a** Representative subcutaneous tumors (left)/orthotopic transplantation tumors (right) collected from Hepa1-6-bearing C57BL/6 mice. **b** Tumor proliferation curves of subcutaneous xenografts in Hepa1-6-bearing C57BL/6 mice. Data presented as mean ± SEM. *n* = 6 mice per group. **c** Kaplan–Meier survival curves of Hepa1-6-bearing C57BL/6 mice. *n* = 6 mice per group. **d** Representative subcutaneous tumors (left)/orthotopic transplantation tumors (right) collected from H22-bearing C57BL/6 mice. **e** Tumor proliferation curves of subcutaneous xenografts in H22-bearing C57BL/6 mice. Data presented as mean ± SEM. *n* = 6 mice per group. **f** Kaplan-Meier survival curves of H22-bearing C57BL/6 mice. *n* = 6 mice per group. **g–i** Flow cytometry analysis of CD8⁺, GZMB⁺CD8⁺, and PD1⁺CD8⁺ in CD3⁺ TILs from Hepa1-6 (**g**) or H22 xenografts (**h**) in C57BL/6 mice and their quantification (**i**). Data presented as mean ± SEM. *n* = 3 mice per group. **j** Representative subcutaneous tumors (left)/orthotopic

transplantation tumors (right) collected from Hepa1-6-bearing BALB/c nude mice. **k** Tumor proliferation curves of subcutaneous xenografts in Hepa1-6-bearing BALB/c nude mice. Data presented as mean ± SEM. *n* = 6 mice per group. **l** Kaplan–Meier survival curves of Hepa1-6-bearing BALB/c nude mice. *n* = 6 mice per group. **m** Representative subcutaneous tumors (left)/orthotopic transplantation tumors (right) collected from H22-bearing BALB/c nude mice. **n** Tumor proliferation curves of subcutaneous xenografts in H22-bearing BALB/c nude mice. Data presented as mean ± SEM. *n* = 6 mice per group. **o** Kaplan–Meier survival curves of H22-bearing BALB/c nude mice. *n* = 6 mice per group. **p, q** T cell-mediated cancer cell-killing assay results. Data presented as mean ± SEM (*n* = 3 independent biological replicates). *P* value was determined by unpaired two-sided Student's *t* test with no correction for multiple comparisons (**b, e, i, k, n, p, q**) and Kaplan-Meier method (**c, f, l, o**). Schematic diagrams (**a, d, j, m**) were created with BioRender.com. Source data are provided as a Source data file.

obvious restraint in the proportions of infiltrated CD8⁺ T cells. However, contrasting results were observed in tumors from the sg*Prdm1* group (Supplementary Fig. 3a–h). Thus, *Prdm1* might promote immune evasion by upregulating PD-L1. To confirm the functional association between *Prdm1*-induced PD-L1 upregulation and in vivo tumor

enlargement, we have also established *Pd-l1*⁻/⁻ Hepa1-6 and H22 cell lines and performed the same in vivo experiment in Fig. 2a, d. The established *Pd-l1*⁻/⁻ Hepa1-6 and H22 cell lines with *Prdm1* over-expression vector and knockout vector, respectively, were inoculated into immune-competent mice (C57BL/6 mice). The results revealed

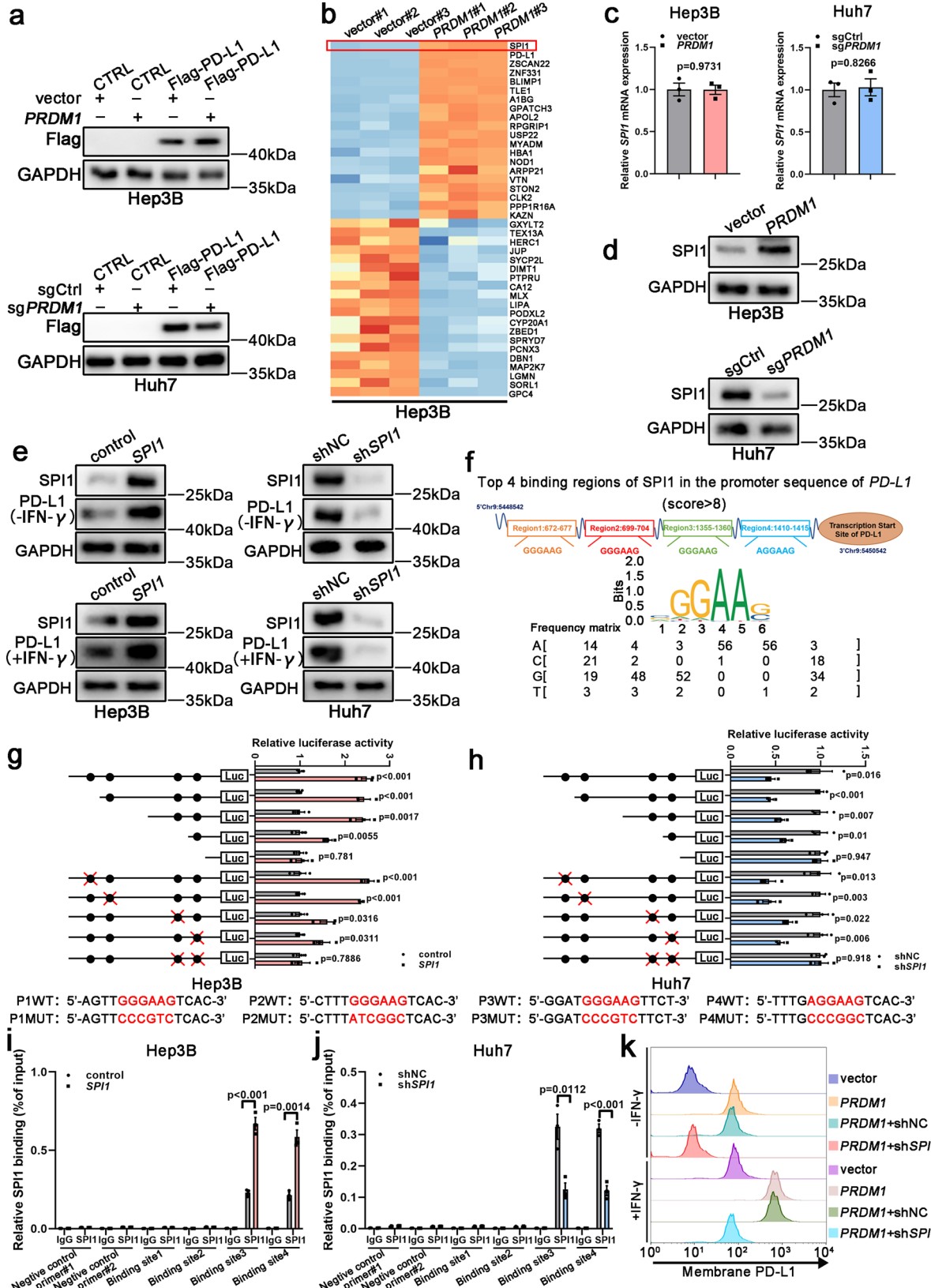

that *Prdm1* overexpression in immune-competent mice inhibited tumor proliferation and that *Prdm1* knockout contributed to tumor proliferation (Supplementary Fig. 3i–n). Thus, above results confirmed that *Prdm1* may promote tumor immune evasion by driving PD-L1 upregulation and neutralizing the anti-tumor efficacy of *Prdm1* overexpression.

## SPI1 serves as a downstream effector of *PRDM1* to enhance *PD-L1* transcription

To further confirm that *PRDM1* enhances *PD-L1* mRNA but not protein expression, CMV-driven Flag-PD-L1-overexpressing HCC cell lines were established. However, PD-L1 levels were comparable in the Flag-PD-L1 cells with or without *PRDM1* overexpression/knockout, suggesting that

**Fig. 3 | SPI1 serves as a downstream effector of PRDM1 to enhance PD-L1 transcription. a** Western blotting of PD-L1/Flag levels in *PRDM1*-overexpressing and vector Hep3B cells or in *PRDM1*-knockout and sgCtrl Huh7 cells. *n* = 3 independent biological replicates. **b** TMT-based quantitative proteomic analysis of Hep3B cells stably overexpressing *PRDM1* and the corresponding control cells, (*PRDM1* and vector cells, respectively), with 3 replicates per group. Heat map showing the top 20 upregulated proteins and the top 20 downregulated proteins between *PRDM1*-overexpressing Hep3B cells and control cells in the proteomics analysis. **c** qRT-PCR of *SPI1* expression in *PRDM1*-overexpressing and vector Hep3B cells or in *PRDM1*-knockout and sgCtrl Huh7 cells. Data presented as mean ± SEM (*n* = 3 independent biological replicates). **d** Western blotting of SPI1 expression in *PRDM1*-overexpressing and vector Hep3B cells or in *PRDM1*-knockout and sgCtrl Huh7 cells. n = 3 independent biological replicates. **e** Western blotting of PD-L1 expression in *SPI1*-overexpressing and control Hep3B cells or *SPI1*-knockdown and shNC Huh7 cells with or without IFN-γ treatment. *n* = 3 independent biological replicates. **f** Putative SPI1-binding sites (SBS) within the genomic sequence adjacent to the transcription start site (TSS) of the *PD-L1* gene. **g**, **h** Luciferase activities of *PD-L1* promoter reporter vectors in Hep3B (**g**) and Huh7 (**h**) cells. Red characters in the binding regions suggest the putative or mutated SPI1-binding sequences. Data presented as mean ± SEM (*n* = 3 independent biological replicates). **i**, **j** ChIP analysis of SPI1 binding to the *PD-L1* promoter in Hep3B (**i**) and Huh7 (**j**) cells. Two promoter regions of *PD-L1* not expected to be bound by SPI1 were employed as negative controls (negative control#1 and #2). Data presented as mean ± SEM (*n* = 3 independent biological replicates). **k** Flow cytometry analysis of PD-L1 expression in Hep3B cells with or without IFN-γ treatment. *n* = 3 independent biological replicates. *P* value was determined by unpaired two-sided Student's *t* test (**c**, **g**, **h**, **i**, **j**). Source data are provided as a Source data file.

*PRDM1* impacts *PD-L1* expression through its endogenous promoter (Fig. 3a and Supplementary Fig. 4a). These data confirmed that *PRDM1*-induced *PD-L1* upregulation occurred at the transcriptional level. To clarify the potential *PRDM1* downstream target regulating *PD-L1* expression, we performed proteomics analysis using Hep3B cells stably overexpressing *PRDM1* and the corresponding control cells (*PRDM1* and vector cells, respectively). SPI1, a transcription factor for *PD-L1*, was the most likely candidate based on fold change (Fig. 3b). qRT-PCR and western blotting assays detected SPI1 expression both at the mRNA and protein levels in HCC cells with *PRDM1* overexpression or knockout, with no obvious differences in *SPI1* mRNA levels in HCC cells with *PRDM1* overexpression or knockout and their corresponding controls. SPI1 protein upregulation and downregulation were confirmed in *PRDM1*-overexpressing and *PRDM1*-knockout HCC cells, respectively (Fig. 3c, d and Supplementary Fig. 4b). *SPI1* overexpression significantly upregulated PD-L1 mRNA and protein levels following incubation with or without IFN-γ, whereas *SPI1* knockdown had the opposite effect (Fig. 3e and Supplementary Fig. 4c). HCC data mining of the GEPIA database indicated a positive relationship between *SPI1* and *PD-L1* expression, and T cell exhaustion markers, providing proof for SPI1-induced *PD-L1* upregulation (Supplementary Fig. 5a–h). Potential SPI1-binding sites (SBS) in the *PD-L1* promoter were predicted using Jaspar. Four putative SBS were observed in the genomic region (NM_017798) (Fig. 3f). Luciferase assays combined with site-deletion or site-directed mutagenesis suggested that SBS 3 and 4 in the *PD-L1* promoter-induced SPI1-enhanced promoter activity in *PRDM1*-overexpressing Hep3B cells. ChIP results also indicated that SPI1 is recruited only to promoter regions containing SBS 3 and 4 in *SPI1*-overexpressing Hep3B cells. Similar results were also obtained in luciferase assays and ChIP assays of Huh7 cells with *SPI1* knockdown (Fig. 3g–j). Thus, binding sites 3 and 4 are critical for activating *PD-L1* transcription by SPI1. Moreover, *PRDM1*-induced PD-L1 upregulation was reversed by *SPI1* knockdown in Hep3B cells with or without IFN-γ stimulation (Fig. 3k). Taken together, SPI1 is a downstream effector of *PRDM1* and directly binds the *PD-L1* promoter to enhance its mRNA expression.

## USP22 inhibits ubiquitin-mediated proteasomal degradation of SPI1

Despite differences in SPI1 protein levels, its mRNA levels were unchanged in HCC cells with differential *PRDM1* expression, suggesting that *PRDM1* may affect SPI1 protein stability (Fig. 3c, d and Supplementary Fig. 4b). To validate this, HCC cells were incubated with the protein synthesis inhibitor, cycloheximide (CHX). *PRDM1* overexpression significantly attenuated SPI1 degradation. In contrast, SPI1 degradation was strikingly facilitated in *PRDM1*-knockout cells (Fig. 4a, b and Supplementary Fig. 4d, e). Thus, we treated sg*PRDM1* or sgCtrl Huh7 cells with the proteasome inhibitor, MG132, and found that MG132 treatment recovered the SPI1 levels in sg*PRDM1* or sgCtrl Huh7 cells; thus, *PRDM1* could modulate SPI1 protein stability through a proteasome-dependent pathway (Fig. 4c and Supplementary Fig. 4f).

The ubiquitin-proteasome system (UPS) enhances proteasomal degradation of its target proteins. We thus examined whether *PRDM1* could modulate SPI1 proteolysis via the UPS. *PRDM1*-overexpressing or *PRDM1*-knockout HCC cells were transiently co-transfected with plasmids encoding HA-tagged ubiquitin. The results revealed that *PRDM1*-overexpressing Hep3B cells showed significantly lower polyubiquitination levels compared to control cells, whereas *PRDM1*-knockout Huh7 cells displayed dramatically higher polyubiquitination levels compared to sgCtrl cells (Fig. 4d, e). These results suggest that *PRDM1* restrains SPI1 polyubiquitination and proteasomal degradation.

Deubiquitinating enzymes (DUBs) control ubiquitin-dependent pathways by separating protein-ubiquitin bonds. To determine the probable DUBs accounting for SPI1 protein stability, we used an efficient and convenient DUB siRNA library, which downregulates the entire DUB family. Among the 98 currently known human DUBs (Supplementary Table 1), downregulation of only two types of DUBs (USP22 and USP33) decreased the SPI1 protein levels by more than 50% (Fig. 4f). We then determined the mRNA and protein expression levels of both USP22 and USP33 in *PRDM1*, vector control, sg*PRDM1*, and sgCtrl HCC cells. *PRDM1* overexpression only increased the mRNA and protein levels of USP22, whereas *PRDM1* knockout decreased the mRNA and protein levels of USP22 (Fig. 4g, h and Supplementary Fig. 4g). Further, following CHX treatment, *USP22* overexpression significantly attenuated SPI1 degradation, which in contrast, was strikingly facilitated in *USP22* knockdown cells (Fig. 4i, j and Supplementary Fig. 4h, i). The *USP22* knockdown-mediated decrease in SPI1 expression was abolished by MG132 (Fig. 4k and Supplementary Fig. 4j). HCC cells with *USP22* overexpression or knockdown were then transiently co-transfected with plasmids encoding HA-tagged ubiquitin. Hep3B cells with *USP22* overexpression showed markedly lower polyubiquitination levels compared to the control cells, whereas *USP22*-knockdown Huh7 cells displayed significantly higher polyubiquitination levels compared to the control cells (Fig. 4l, m). Thus, *PRDM1* was inferred to stabilize SPI1 via USP22.

## USP22 interacts with SPI1

The immunoprecipitation and mass spectrometry (IP/MS) results validated USP22 as an SPI1-interacting protein (Supplementary Fig. 6a, b). Co-IP assays were then performed to verify the interaction between USP22 and SPI1. USP22 was immunoprecipitated with an anti-SPI1 antibody and vice versa (Fig. 5a). Immunofluorescence (IF) assays also revealed the co-localization of USP22 and SPI1 (Fig. 5b). USP22 contains an N-terminal zinc finger domain and a C19 ubiquitin-specific peptidase domain (Fig. 5c). To identify the region within USP22 interacting with SPI1, we generated two truncated mutants of USP22 in HEK-293T cells. The C-terminal ubiquitin-specific peptidase domain of USP22 binds SPI1 strongly, whereas the N-terminal zinc finger domain binds SPI1 weakly (Fig. 5d). Moreover, the N-terminal zinc finger domain of USP22 was not involved in SPI1 ubiquitination (Fig. 5e). Point mutants of USP22 were also constructed in HEK-293T cells (Fig. 5f). In line with

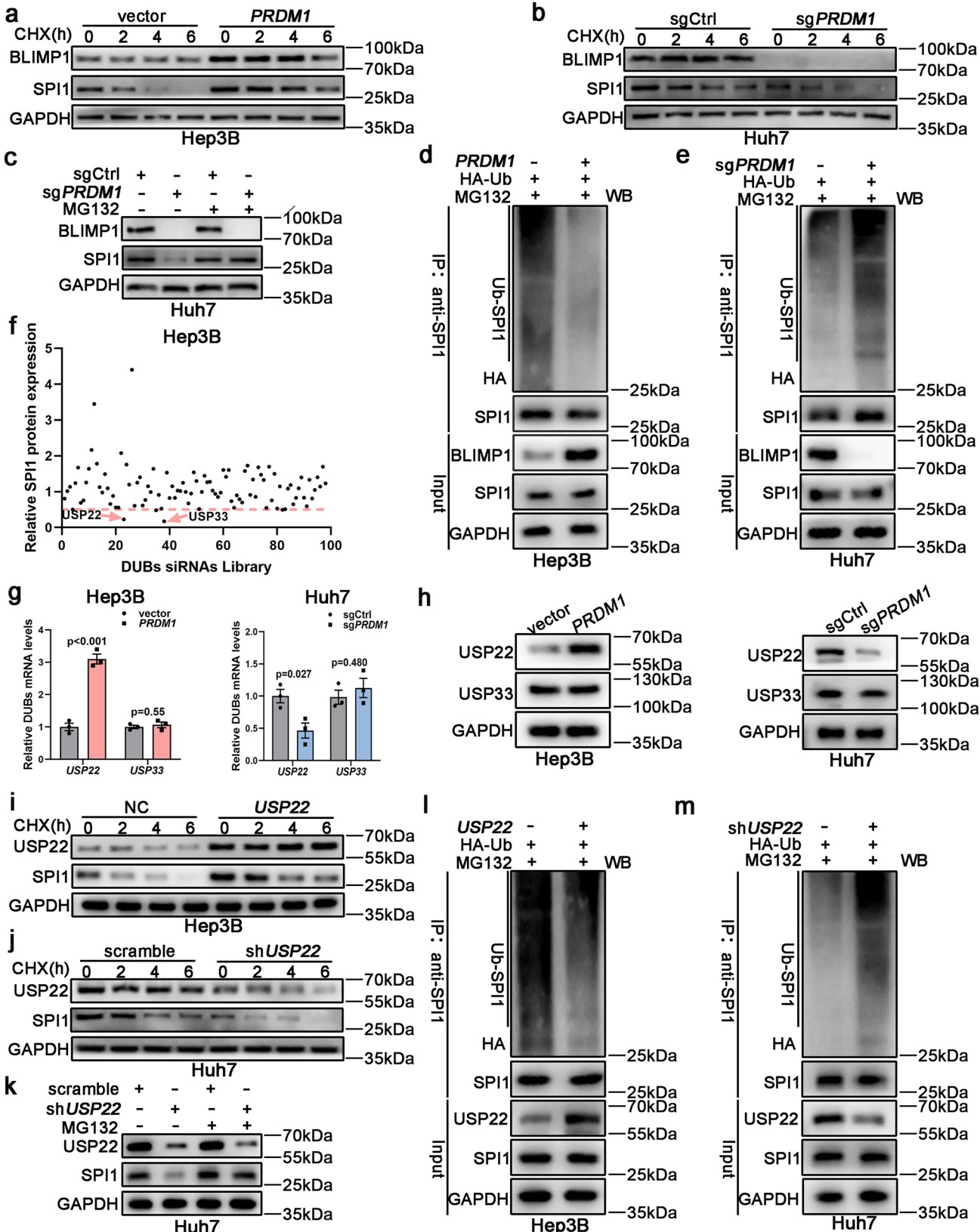

the truncated mutant of USP22, the C-terminal ubiquitin-specific peptidase domain of USP22 containing a point mutation could bind SPI1 weakly, whereas the N-terminal zinc finger domain of USP22 containing a point mutation could bind SPI1 more strongly (Fig. 5g). Further, the C-terminus of USP22 containing a point mutation did not affect SPI1 ubiquitination (Fig. 5h). The C-terminal ubiquitin-specific

peptidase domain of USP22 containing a point mutation did not affect the degradation of SPI1 following CHX treatment (Fig. 5i). Thus, the C-terminal ubiquitin-specific peptidase domain of USP22 is the dominant regulator of binding between USP22 and SPI1. To determine which domain of SPI1 was responsible for its binding to USP22, six truncated SPI1 proteins were generated in HEK-293T cells (Fig. 5j). Only

**Fig. 4 | USP22 inhibits ubiquitin-proteasomal degradation of SPI1. a, b** SPI1 protein expression in *PRDM1*-overexpressing and vector Hep3B cells (**a**) or in *PRDM1*-knockout and sgCtrl Huh7 cells (**b**). *n* = 3 independent biological replicates. **c** SPI1 protein expression was determined in *PRDM1* knockout and sgCtrl Huh7 cells. *n* = 3 independent biological replicates. **d** SPI1 was pulled down and ubiquitin-conjugated SPI1 was then determined by immunoblotting using an anti-HA antibody. *n* = 3 independent biological replicates. **e** SPI1 was pulled down and ubiquitin-conjugated SPI1 was determined by immunoblotting using an anti-HA antibody. *n* = 3 independent biological replicates. **f** DUB siRNA library screening indicated that USP22 and USP33 suppression decreased SPI1 protein levels. **g** qRT-PCR of *USP22* or *USP33* expression in *PRDM1*-overexpressing and vector Hep3B cells or in *PRDM1*-knockout and sgCtrl Huh7 cells. Data presented as mean ± SEM (*n* = 3 independent biological replicates). **h** Western blotting of USP22 or USP33 expression in *PRDM1*-overexpressing and vector Hep3B cells or in *PRDM1*-knockout and sgCtrl Huh7 cells. *n* = 3 independent biological replicates. **i, j** SPI1 protein expression in *USP22*-overexpressing and vector Hep3B cells (**i**) or in *USP22*-knockdown and control Huh7 cells (**j**). *n* = 3 independent biological replicates. **k** SPI1 protein expression was determined in *USP22*-knockdown or control Huh7 cells. *n* = 3 independent biological replicates. **l** SPI1 was pulled down and ubiquitin-conjugated SPI1 was then determined by immunoblotting using an anti-HA antibody. *n* = 3 independent biological replicates. **m** SPI1 was pulled down and ubiquitin-conjugated SPI1 was then determined by immunoblotting using an anti-HA antibody. *n* = 3 independent biological replicates. *P* value was determined by unpaired two-sided Student's *t* test (**g**). Source data are provided as a Source data file.

SPI1 mutants containing the C-terminal (aa 170–270) region, but not the N-terminal region (aa 1–85) and the middle region (aa 86–169), were found to interact with USP22 (Fig. 5k). Additionally, Myc-tagged USP22 (the C-terminus ubiquitin-specific peptidase) and His-tagged SPI1 D3 were mutually immunoprecipitated (Fig. 5l). These results indicate that USP22 interacts with SPI1 and enhances SPI1 stability through deubiquitination.

### PRDM1/BLIMP1 regulates *USP22* transcription
Based on the above results, *PRDM1* overexpression could upregulate both USP22 mRNA and protein expression. We thus wondered whether *PRDM1*/BLIMP1 regulates *USP22* expression transcriptionally. Intriguingly, a potential BLIMP1-binding site (BBS) in the *USP22* promoter was determined using Jaspar (Fig. 5m). The luciferase assay indicated that *PRDM1* overexpression induced higher promoter activity in Hep3B cells, whereas the opposite effect was observed in *PRDM1*-knockout Huh7 cells. Further, site-directed mutagenesis abolished the increased promoter activity in Hep3B cells with *PRDM1* overexpression and the decreased promoter activity in *PRDM1*-knockout Huh7 cells (Fig. 5n, o). ChIP assays validated BLIMP1 occupancy at the *USP22* promoters in Hep3B and Huh7 cells (Fig. 5p, q).

### *Prdm1* overexpression increases the efficacy of PD-1 mAb therapy in mice
We then examined whether *Prdm1* overexpression could enhance the sensitivity of PD-1 mAb therapy. Thus, the IgG isotype or PD-1 mAb was employed to treat C57BL/6 mice inoculated with *Prdm1* overexpressing or knockout cells and their corresponding control cells. The *Prdm1* overexpression group showed no visible difference in tumor proliferation and survival time compared with the control group. Similarly, the *Prdm1* knockout group showed similar tumor proliferation and survival time comparable to the sgCtrl group (Fig. 6a–h). Nevertheless, PD-1 mAb treatment dramatically inhibited tumor proliferation and extended survival time compared to the IgG group in Hepa1-6 cells. More importantly, combined treatment with *Prdm1* overexpression and PD-1 mAb further impaired tumor proliferation and prolonged survival time compared with that in *Prdm1* overexpression or PD-1 mAb treatment alone (Fig. 6a, c, e, f). Tumors were harvested for further analysis at the end of the treatment. Flow cytometry analysis and IF staining revealed that co-treatment with *Prdm1* overexpression and PD-1 mAb significantly increased the infiltration of CD8+ T cells and the activity (GZMB+) of infiltrated CD8+ T cells, and reduced the exhaustion of infiltrated CD8+ T cells as reflected by PD-1+ staining compared with that in the *Prdm1* overexpression group (Fig. 6i and Supplementary Fig. 7a–c). However, co-treatment with *Prdm1* knockout and PD-1 mAb had a limited effect on tumor growth and survival time compared with *Prdm1* knockout alone (Fig. 6b, d, g, h). Flow cytometry analysis and IF staining confirmed that co-treatment with *Prdm1* knockout and PD-1 displayed comparable percentages of infiltrated CD8+ T cells, activities (GZMB+) of infiltrated

CD8+ T cells, and exhaustion of infiltrated CD8+ T cells compared with those in *Prdm1* knockout alone (Fig. 6j and Supplementary Fig. 7d–f). Moreover, immunocompetent orthotopic models also confirmed that co-treatment with *Prdm1* overexpression and PD-1 mAb further impaired tumor proliferation compared with that in *Prdm1* overexpression or PD-1 mAb treatment alone, whereas co-treatment with *Prdm1* knockout and PD-1 mAb had a limited effect on tumor growth compared with that in *Prdm1* knockout alone (Fig. 6c, d).

A mouse model in which HCC was induced by a combination of diethylnitrosamine (DEN) and repeated carbon tetrachloride (CCl4) treatment was also employed. At 22 weeks, mice were treated with PD-1 mAb and an adeno-associated virus serotype 8 (AAV8) system to generate mice overexpressing *Prdm1* in the liver via tail vein injection of AAV8-GFP-*Prdm1* or AAV8-GFP-control virus driven by a liver-specific promoter (thyroxine-binding globulin) (Fig. 6k). In this model, we further validated that combined treatment could result in less advanced liver lesions, as assessed by gross appearance (Fig. 6l), H&E staining (Fig. 6m), tumor numbers (Fig. 6n), and maximum tumor diameter (Fig. 6o). Meanwhile, the combined treatment maximized the survival of HCC-bearing mice (Fig. 6p). Collectively, tumoral *Prdm1* expression in HCC might be used as a predictive biomarker for superior immunotherapeutic efficacy.

To further confirm that the synergistic effect of co-treatment with *Prdm1* overexpression and PD-1 mAb was dependent on CD8+ T cells, CD8α mAb was used for the in vivo experiments. We found that CD8α mAb cotreatment aggravated tumor burden by eliminating CD8+ T cells in both Hepa1-6 and H22 subcutaneous and orthotopic tumor models (Supplementary Fig. 8a–h). Thus, CD8+ T cells are indispensable for the synergistic effect of co-treatment with *Prdm1* overexpression and PD-1 mAb. Collectively, these data confirm that *Prdm1* overexpression shapes an immunosuppressive microenvironment by upregulating PD-L1 expression, thus contributing to the increased efficacy of PD-1 mAb therapy.

### PRDM1-USP22-SPI1 axis regulates PD-L1 levels in patients with HCC
To confirm our hypothesis in HCC tissues, we examined the protein levels of BLIMP1, USP22, SPI1, PD-L1, and CD8α using a series of tissue microarrays (TMA) in cohort 2, which contained 90 patients with HCC. Our TMA results revealed that the protein levels of BLIMP1, USP22, and SPI1 were positively associated with the PD-L1 protein levels and were negatively associated with CD8+ T cell infiltration in HCC tissues (Fig. 7a, b). We also investigated the protein levels of BLIMP1, USP22, SPI1, PD-L1, and CD8α by multi-color immunohistochemistry (IHC) in cohort 2. Multi-color IHC revealed that BLIMP1, USP22, and SPI1 levels had a positive relationship with PD-L1 expression (Fig. 7c). Moreover, BLIMP1 and PD-L1 levels were negatively correlated with CD8+ T cells, and the activity (GZMB+) of infiltrated CD8+ T cells (Fig. 7d). These data were in accordance with the in vitro and in vivo findings.

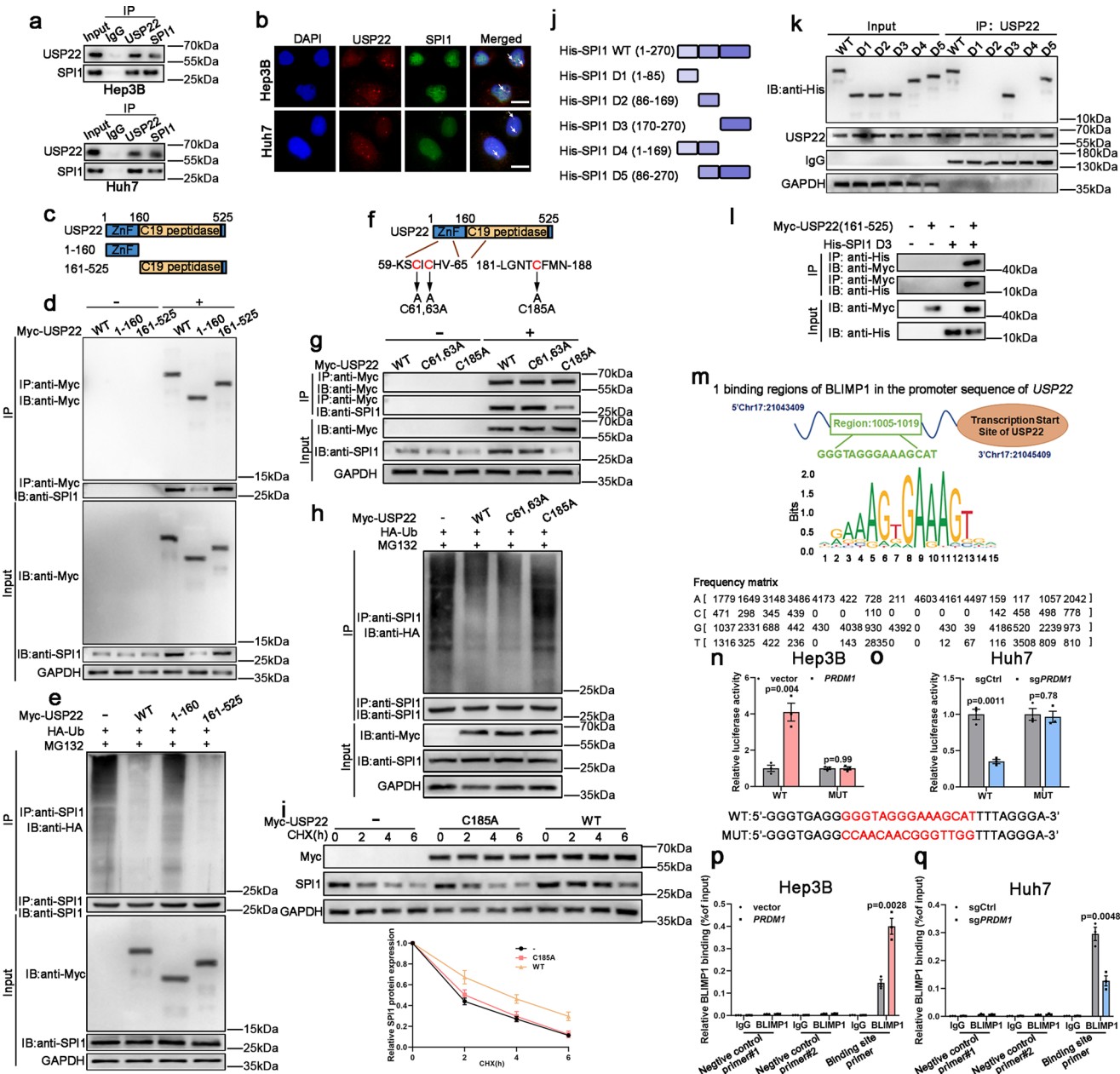

**Fig. 5 | USP22 was identified as a SPI1-interacting protein. a** Co-IP experiments indicated the interaction of endogenous USP22 and SPI1. *n* = 3 independent biological replicates. **b** Confocal microscopy showing colocalization of USP22 (red) with SPI1 (green). *n* = 3 independent biological replicates. Scale bars, 20 μm. **c** Schematic illustration of USP22 and its mutants. **d** USP22 and its mutants were immunoprecipitated and the bound SPI1 was determined. *n* = 3 independent biological replicates. **e** SPI1 ubiquitination was determined by SPI1 immunoprecipitation and western blotting using an anti-HA antibody. *n* = 3 independent biological replicates. **f** Schematic illustration of USP22 and its point mutants. **g** USP22 and its point mutants were immunoprecipitated and the bound SPI1 was determined. *n* = 3 independent biological replicates. **h** The effects of USP22 and its point mutants on SPI1 ubiquitination were confirmed. *n* = 3 independent biological replicates. **i** The protein levels of USP22 (Myc) and SPI1 were detected. Data presented as mean ± SEM (*n* = 3 independent biological replicates). **j** Schematic representation of full-

length SPI1 and truncated SPI1. **k** Interactions between USP22 and full-length or truncated SPI1 were analyzed using co-IP in HEK-293T cells. *n* = 3 independent biological replicates. **l** Co-IP experiments indicated the interaction of Myc-USP22 (161-525) and His-SPI1 (D3) in HEK-293T cells. *n* = 3 independent biological replicates. **m** Putative BLIMP1-binding site (BBS) within the genomic sequence adjacent to TSS of *USP22* gene. **n, o** Luciferase activities of *USP22* promoter reporter vectors in Hep3B (**n**) and Huh7 (**o**) cells. Red characters in the binding regions suggest putative or mutated BLIMP1 binding sequences. Data presented as mean ± SEM (*n* = 3 independent biological replicates). **p, q** ChIP analysis of BLIMP1 binding to the *USP22* promoter in Hep3B (**p**) and Huh7 (**q**) cells. Two promoter regions of *USP22* not expected to be bound by BLIMP1 were employed as negative controls. Data presented as mean ± SEM (*n* = 3 independent biological replicates). *P* value was determined by unpaired two-sided Student's *t* test (**n, o, p, q**). Source data are provided as a Source data file.

## Single-cell analysis of intra-tumoral immune cell populations confirmed that *PRDM1* overexpression potentiates T cell exhaustion

Single-cell RNA-seq (scRNA-seq) has been validated as a tool to dissect tumor heterogeneity and to assess tumor microenvironment interactions. Hence, to show an unbiased and comprehensive perspective of

the tumor microenvironment affected by *PRDM1* expression, we performed scRNA-seq using HCC biopsies before PD-1 mAb-based therapies. In total, 8277 cells from two patients who passed the quantity control were enrolled in the subsequent analysis. Based on canonical markers, the cells exhibited eight distinct types, and the UMAP plot revealed subcluster distribution among the two patients (Fig. 8a).

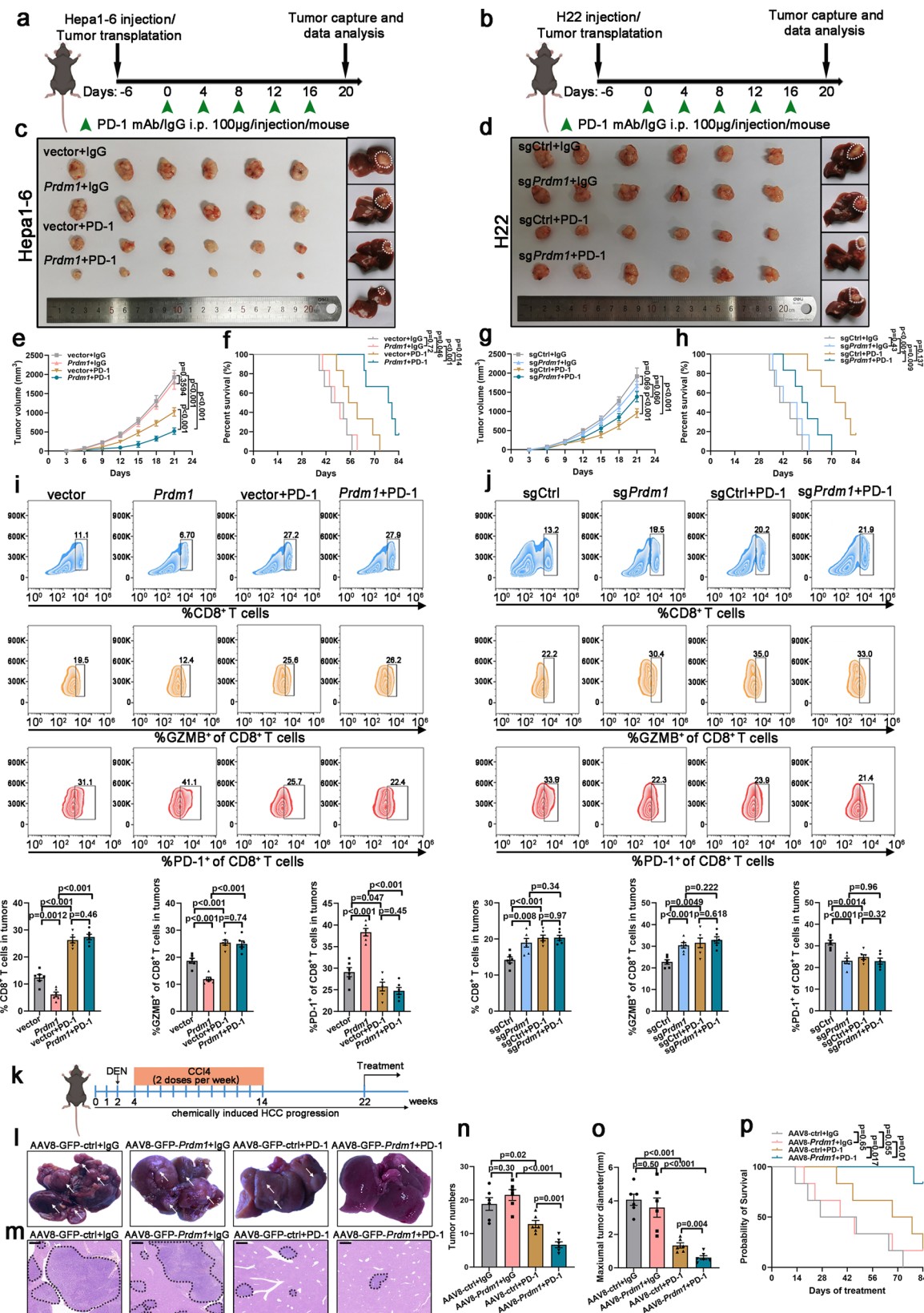

Figure 8b shows the highly expressed genes across the eight cell types. We then reclustered the tumor cells into two sub-clusters. The sub-clusters indicated that patient 2 expressed higher levels of *PRDM1* (Fig. 8c). To investigate the effects of *PRDM1* on the transcriptome of tumor-infiltrating T cells, we conducted unbiased secondary clustering of the T cell population. In this study, T cells were divided into nine

distinct subclusters. According to the top 10 DEGs, they were designated as C0-CD4-IL7R, C1-CD8-GZMK, C2-CD4-FOXP3, C3-CD8-GNLY, C4-CD8-doublets, C5-CD8-TOP2A, C6-CD8-KLRB1, C7-CD8-LAG3, C8-CD8-TRBV15, and their violin plots showed the expression of marker genes in each T cell cluster (Fig. 8d). Among them, C1, C3, and C6 were thought to be cytotoxic T cells. Next, we compared the abundance of

**Fig. 6 | Synergistic effect of Prdm1 overexpression and PD-1 mAb therapy in mice. a, b** Schematic view of the treatment plan in subcutaneous tumors and orthotopic tumors. C57BL/6 mice were implanted with Hepa1-6/H22 cells subcutaneously or as orthotopic tumors and were treated with PD-1 mAb or IgG isotype control. **c, d** Representative xenograft tumors (left) and orthotopic tumors (right) obtained after euthanizing the mice. **e, f** Tumor proliferation curves (**e**) and Kaplan–Meier survival curves (**f**) of Hepa1-6-bearing C57BL/6 mice. Data presented as mean ± SEM (**e**). n = 6 mice per group. **g, h** Tumor proliferation curves (**g**) and Kaplan-Meier survival curves (**h**) of H22-bearing C57BL/6 mice. Data presented as mean ± SEM (**g**). n = 6 mice per group. **i, j** Flow cytometry analysis of CD8$^+$, GZMB $^+$CD8$^+$, and PD1$^+$CD8$^+$ in CD3$^+$ TILs from Hepa1-6 (**i**) or H22 (**j**) subcutaneous tumors in C57BL/6 mice. Data presented as mean ± SEM. n = 6 mice per group. **k** A schematic view of the treatment plan in the DEN/CCL4-induced HCC model. **l** Representative images of liver tumors (annotated by white arrows). **m** H&E staining of liver sections of sacrificed mice. Scale bars, 400 μm. **n, o** Quantification of tumor numbers (**n**) and maximal tumor sizes (**o**). Data presented as mean ± SEM. n = 6 mice per group. **p** Kaplan–Meier survival curves in the DEN-induced HCC model. n = 6 mice per group. P value was determined by unpaired two-sided Student's t test (**e, g, i, j, n, o**) and Kaplan–Meier method (**f, h, p**) with no correction for multiple comparisons. Schematic diagrams (**a, b, k**) were created with BioRender.com. Source data are provided as a Source data file.

each subpopulation between patients 1 and 2. We noticed that CD8$^+$ cytotoxic T cell subpopulations were predominant in patient 1, who showed low expression of *PRDM1* in tumor cells, whereas patient 2 showed a higher percentage of CD8$^+$LAG3$^+$ exhausted T cells and CD4$^+$FOXP3$^+$ Tregs, supporting the notion that *PRDM1* overexpression potentiates T cell exhaustion (Fig. 8e). We further conducted Gene Ontology (GO) analysis of the DEGs in tumor cells from patient 2. Immune regulatory parameters such as positive regulation of immune response, response to interferon-gamma, and lymphocyte activation, were significantly enriched. Regulation of the cell death pathway was also significantly enriched, indicating that *PRDM1* might suppress the proliferation ability of HCC cell lines by mediating tumor cell death (Fig. 8f). Ultimately, as expected, patient 2 with high tumoral *PRDM1* expression showed significant tumor shrinkage after treatment, whereas patient 1 with low tumoral *PRDM1* expression showed tumor progression after treatment.

Meanwhile, to further validate the results of scRNA-seq, we have increased the number of HCC biopsies before PD-1 mAb-based therapies and performed multi-color IHC. As expected, BLIMP1 protein levels were negatively correlated with CD8$^+$ T cells and the activity (GZMB$^+$) of infiltrated CD8$^+$ T cells (Supplementary Fig. 9a, b). Meanwhile, patients with high tumoral BLIMP1 protein expression showed significant tumor shrinkage after treatment, whereas patients with low tumoral BLIMP1 protein expression showed tumor progression after treatment (Supplementary Fig. 9c, d). The results confirmed our hypothesis that *PRDM1* overexpression contributes to the therapeutic effects of PD-1 mAb therapy.

## Discussion

As a vital factor that markedly influences the outcome of PD-(L)1 checkpoint blockade, modulation of PD-L1 expression has been studied extensively. Various critical transcription factors, including HIF-1α, c-MYC, NF-kB, STAT3, c-JUN, and miR-138-5p have been shown to transcriptionally regulate PD-L1 expression[25–30]. Moreover, several critical proteins including CMTM4, CMTM6, GSK3β, CSN5, CDK4, CDK6, and palmitoylated B3GNT3 have been shown to affect post-translational PD-L1 stability[31–36]. In our study, based on screening using the online tool GEPIA and establishment of CMV-driven Flag-PD-L1 overexpressing HCC cell lines, we identified that *PRDM1* enhances the transcription of *PD-L1*. Functionally, T cell-mediated cancer cell-killing assays and subcutaneous and orthotopic models demonstrated that *PRDM1* overexpression in HCC cells dampens T cell-induced cytotoxicity by upregulating tumoral PD-L1 expression. Proteomics analysis revealed that SPI1 may serve as a pivotal transcriptional factor that enhances PD-L1 expression by acting as a downstream effector of *PRDM1*. Previous studies have revealed that SPI1 upregulation suppresses tumor growth in MYC-deregulated B cell lymphomas, acute myeloid leukemia, and leukemia. In contrast, SPI1 has been reported to exert an oncogenic role in non-small cell lung carcinoma (NSCLC) and virally induced murine erythroleukemia. Nevertheless, its role in tumor immunity remains unclear. Herein, we identified that *PRDM1* may upregulate SPI1 expression, thus contributing to PD-L1 expression. Ubiquitination is rigorously regulated by ubiquitin ligases or deubiquitinating enzymes. DUBs are proteases that modulate ubiquitin-

dependent pathways by cleaving ubiquitin-protein bonds. Here, by utilizing the DUB siRNA library, we found that USP22 could reduce SPI1 protein degradation through deubiquitination. Supporting this view, a recent study revealed that USP22 could regulate PD-L1 expression in a direct and indirect way[37]. Herein, we found that USP22-induced deubiquitination of SPI1 dominantly enhanced *PD-L1* transcription. Moreover, E3 ligases are also responsible for protein degradation. We could not fully exclude the possibility that *PRDM1* may also influence SPI1 protein stability through a specific E3 ligase. Interestingly, BLIMP1 was then identified as a vital transcription factor of *USP22* in our study. Thus, this study uncovers a regulatory mechanism for tumoral PD-L1 expression through the PRDM1-USP22-SPI1 axis.

PD-(L)1 mAb has been proven to be beneficial for patients with various advanced malignancies. However, only a fraction of patients with HCC respond to PD-(L)1 mAb therapy. Therefore, the application of optional strategies such as combination therapy has been extensively explored. Here, we found that *PRDM1*-overexpressing HCC cells upregulated PD-L1 expression and inhibited T cell-mediated antitumor immunity, thus dampening its role as a tumor growth suppressor in immunocompetent mice. We thus propose that PD-(L)1 blockade conquers HCC resistance to *PRDM1* overexpression by overcoming immune surveillance. Our preclinical animal experiments including subcutaneous tumor models, orthotopic tumor models, and the DEN-induced HCC model, validated the synergistic effect of *Prdm1* overexpression and PD-1 mAb treatment, in terms of a distinct reduction in tumor size/number and extended overall survival, which was attributed to the reinforcement of TILs in the tumors. In HCC samples, we determined that BLIMP1, USP22, and SPI1 were positively associated with PD-L1 expression. Therefore, we propose a combination therapeutic strategy for treating patients with HCC. Further, ideal therapeutic vectors (e.g., adeno-associated virus) carrying *PRDM1* gene are currently needed to be developed to evaluate their safety and potential to serve as vital anti-tumor drugs, for synergistically increasing the efficacy of PD-(L)1-based therapies. Moreover, the underlying mechanism that separates the two presumed effects of *PRDM1*/BLIMP1 needs to be investigated further.

In summary, we identified that tumoral *PRDM1*/BLIMP1 overexpression is a double-edged sword in regulating tumor growth. *PRDM1*/BLIMP1 overexpression inhibits cell-intrinsic cell growth while promoting tumor cell immune evasion by up-regulating PD-L1 and dampening CD8$^+$ T cell anti-tumor immune response simultaneously. Furthermore, *PRDM1*/BLIMP1 overexpression combined with PD-(L)1 mAb treatment provides a therapeutic strategy for the treatment of patients with HCC (Fig. 7e).

## Methods
### Clinical tissue samples
Three independent cohorts of HCC patients were collected from the Hepatobiliary Center of The First Affiliated Hospital of Nanjing Medical University. Cohort 1 containing 40 fresh HCC tissues was obtained from patients treated between January 2020 and May 2020. Cohort 2 containing 90 paired HCC and adjacent normal tissues was acquired from patients treated between February 2013 and October 2015. No

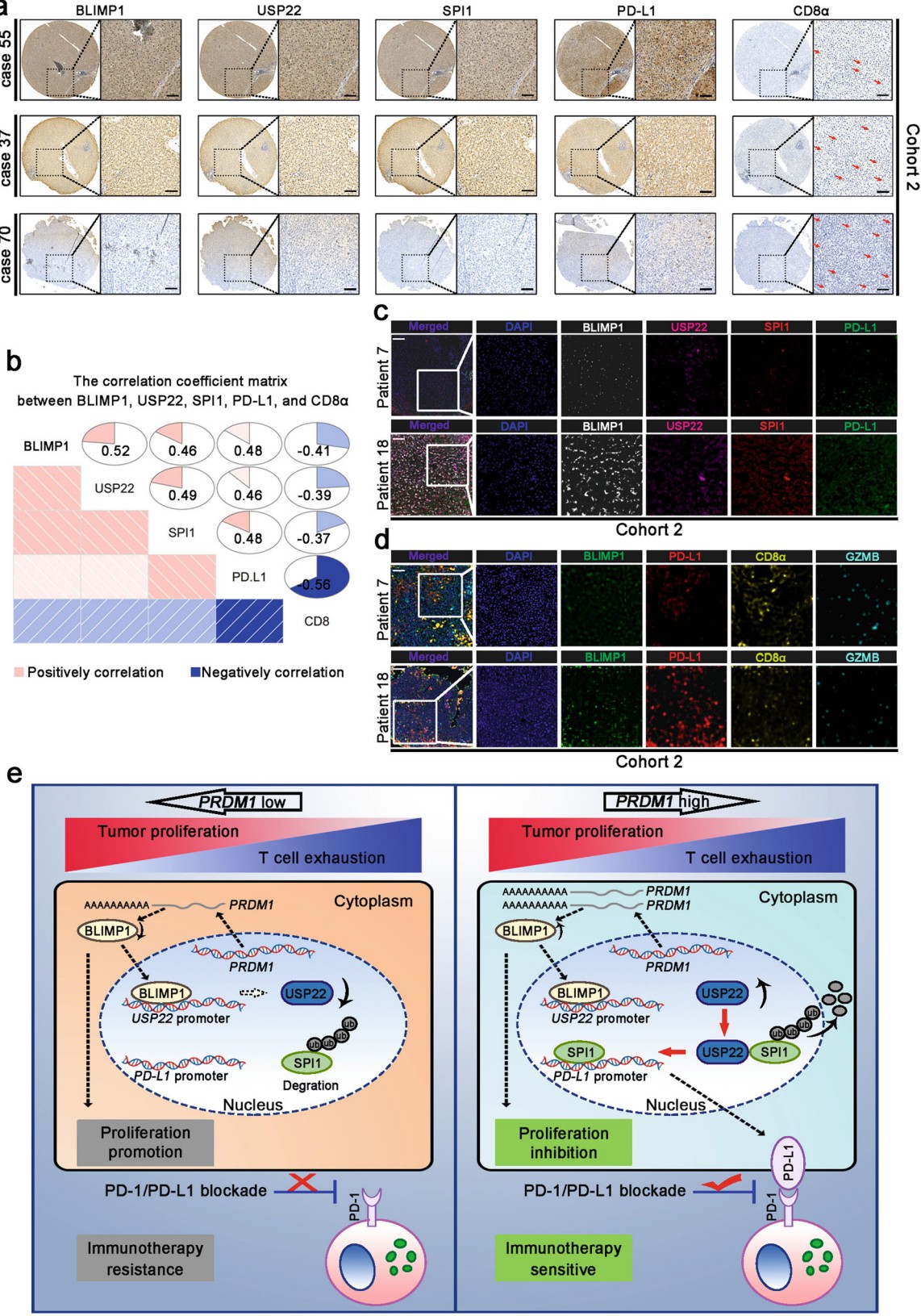

**Fig. 7 | The PRDM1-USP22-SPI1 axis regulates PD-L1 levels in patients with HCC.**
**a**, **b** Immunohistochemical staining of BLIMP1, USP22, SPI1, PD-L1, and CD8α expression in patients with HCC. $n$ = 90 patients. Scale bars, 100 mm. **c**, **d** Multi-color immunohistochemistry using BLIMP1, USP22, SPI1, and PD-L1 (**c**) or BLIMP1, PD-L1, CD8α, and GZMB (**d**) antibodies in patients with HCC. $n$ = 90 patients. Scale bars, 100 μm. **e** Proposed model underlying the roles of *PRDM1*/BLIMP1 in promoting tumor immune evasion in HCC. A schematic diagram was designed using BioRender.

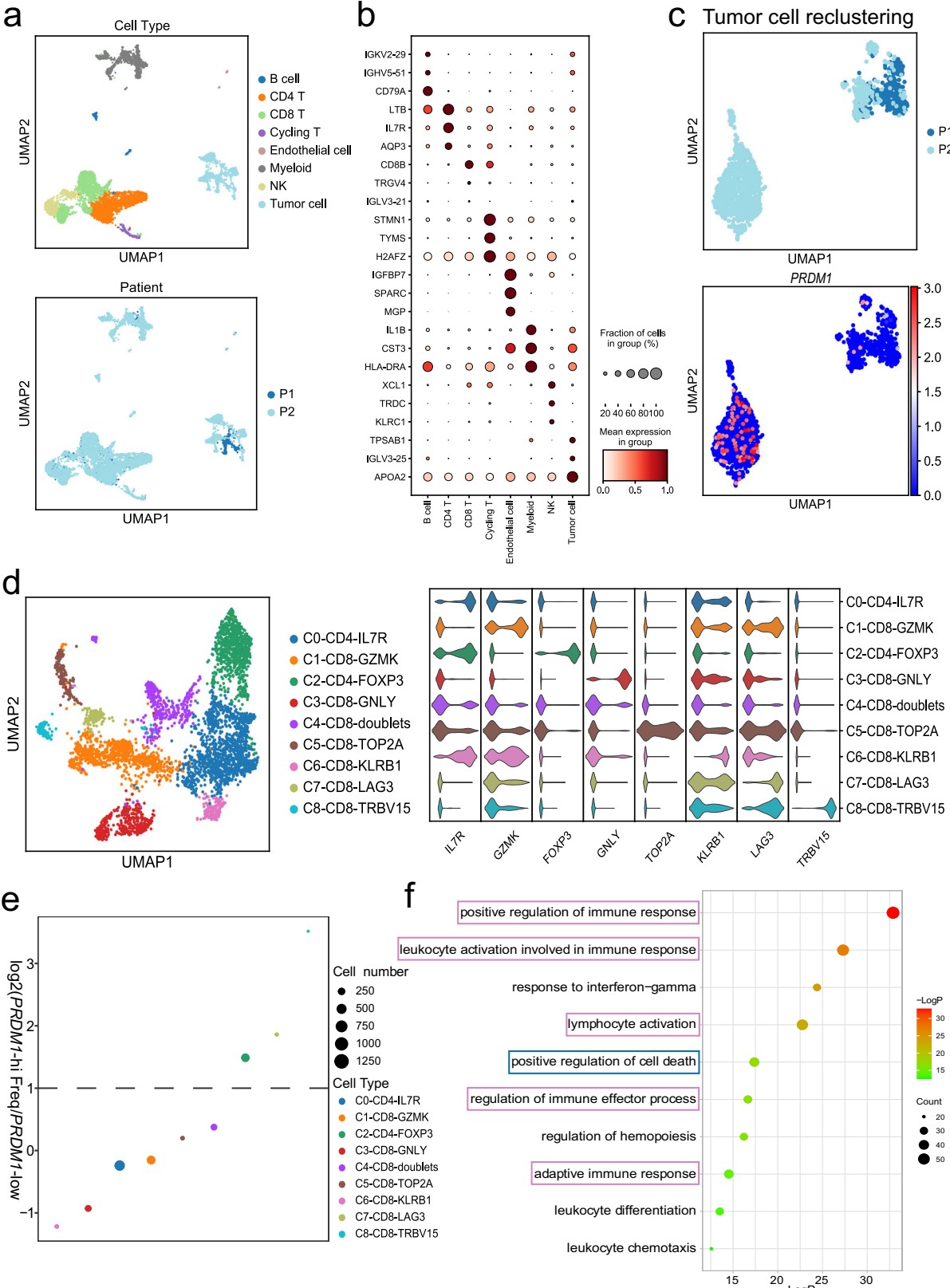

**Fig. 8 | Single-cell analysis of intra-tumoral cell populations. a** In total, 8 cell types were identified and shown using UMAP. The colors were coded by cell types (top) and patients (bottom). **b** Dot plot showing the top marker genes of each cell type. **c** UMAP plots (top) and *PRDM1* expression (bottom) in malignant cells from two patients. **d** UMAP plot and violin plot showing distinct T cell subclusters and their marker genes. **e** Dot plot depicting the relative frequency of each T cell subpopulation between patient 1 and patient 2. **f** The top 10 GO terms of upregulated genes in *PRDM1*-high tumor cells from patient 2.

patient in Cohort 1 and Cohort 2 received any other HCC treatment prior to surgery. Cohort 3 containing 22 HCC patients' biopsies before PD-1 mAb-based therapies that responded or did not respond to Camrelizumab (3 mg/kg, q2w) were also collected from the Hepatobiliary Center of The First Affiliated Hospital of Nanjing Medical University. The data of their clinicopathological features were anonymized and shown in Supplementary Table 2. Each specimen was histologically and pathologically examined and graded by two experienced pathologists. Our study was approved by the Ethics Committee of the First Affiliated Hospital of Nanjing Medical University, and informed consent was obtained from each patient.

## Cell culture and reagents
Hep3B (HB-8064), Hepa1-6 (CRL-1830), and 293 T (CRL-3216) cell lines were purchased from the American Type Culture Collection (ATCC). Huh7 cell line (JCRB0403) was purchased from the Japanese Cancer Research Resources Bank (JCRB). H22 cell line (GDC0091) was purchased from China Center for Type Culture Collection (CCTCC). Cells were routinely cultured in Dulbecco's modified Eagle medium (DMEM) containing 10% fetal bovine serum (FBS) (Wisent, Nanjing, China) at 37 °C. MG132 (S2619) and cycloheximide (CHX) were purchased from Sigma-Aldrich.

## Plasmids and vectors
To generate stably transfected cell lines, *PRDM1/Prdm1*, *USP22*, sh*USP22*, *SPI1*, and sh*SPI1* lentiviral vectors were purchased from Genechem (Shanghai, China). Lentiviral sg*PRDM1*/sg*Prdm1* vectors using CRISPR/Cas9 gene editing were employed to obtain *PRDM1/Prdm1* knockout HCC cell lines. Sequences of oligonucleotides for sgRNA/shRNA were shown in Supplementary Table 3. For transient reporter analyses, all promoter plasmids and site deletion or site-directed mutagenesis plasmids were synthesized by Genechem (Shanghai, China).

## Flow cytometry
Freshly resected mouse tumors or tumors from patients with HCC of comparable size (-100 mm$^3$) were dispersed into single-cell suspensions. The cells were incubated with the appropriate antibodies at room temperature for 30 min. After washing twice in PBS, the samples were analyzed using FlowJo V10.4 (Beckman Coulter, USA). Antibodies used were: human CD3 (14-0037-82, Thermofisher, 1:200), mouse CD3 (14-0032-82, Thermofisher, 1:200), human CD8 (17-0088-42, Thermofisher, 1:200), mouse CD8 (17-0081-82, Thermofisher, 1:200), human TNF alpha (48-7349-42, Thermofisher, 1:200), human Granzyme B (48-8896-42, Thermofisher, 1:200), mouse Granzyme B (48-8898-82, Thermofisher, 1:200), human PD-1 (12-9969-42, Thermofisher, 1:200), and mouse PD-1 (12-9985-82, Thermofisher, 1:200). Gating strategies used for flow cytometry staining was provided in Supplementary Fig. 10.

## Quantitative real-time PCR (qRT-PCR)
Total RNA was extracted from HCC cells and tumor tissues using TRIzol reagent (Invitrogen). RNA quality and concentration were analyzed using Nanodrop 2000. The extracted RNA was reverse transcribed to cDNA using the TransScript® II First-Strand cDNA Synthesis SuperMix (Transgen). qRT-PCR was then performed using SYBR Green PCR Master Mix (Yeasen). The primers used are listed in Supplementary Table 4.

## Western blotting
The collected cells were lysed using a solution containing RIPA lysis buffer, phosphatase inhibitors, and protease inhibitors (Beyotime, China). The BCA reagent (Beyotime, China) was used to measure protein concentrations. Commensurable amounts of protein were separated using SDS-PAGE, transferred to a membrane, and incubated with various antibodies. Finally, the data were acquired using image Lab 5.2.1. Antibodies used were: BLIMP1 (ab243146, Abcam, 1:1000), USP22 (ab195289, Abcam, 1:1000), USP33 (ab237510, Abcam, 1:1000),

SPI1 (ab227835, Abcam, 1:1000), PD-L1 (ab205921, Abcam, 1:1000), GAPDH (#5174, Cell Signaling Technology, 1:1000), HRP-linked anti-rabbit IgG (#7074, Cell Signaling Technology, 1:3000), and HRP-linked anti-mouse IgG (#7076, Cell Signaling Technology, 1:3000). Uncropped and unprocessed scans of blots are included in a Source Data file.

## T cell-mediated tumor cell killing assay
To obtain activated T cells, peripheral blood mononuclear cells (PBMCs) from healthy donors were cultured in CTS™ AIIM V™ SFM (Gibco) containing 1000 U/mL recombinant human IL-2 (R&D) and human CD3/CD28/CD2 T cell activator (Stemcell Technologies) for seven days. The experiments were carried out with 100 ng/mL anti-CD3 antibody and 1000 U/mL IL-2. HCC cells were seeded in plates, incubated overnight, and then co-cultured with activated T cells for 2 days at the HCC cell: T cell ratio of 1:3. The plates were then washed with PBS to remove cell debris and T cells. The remaining living HCC cells were stained with crystal violet and analyzed using a spectrometer at 570 nm.

## PD-L1 and PD1 interaction assay
HCC cells were plated in 12-well plates and incubated with IFN-γ (500 IU/ml, 24 h). These cells were then treated with FITC-labeled PD-1. PD-1 binding was detected and quantified every 2 h.

## Tandem mass tag (TMT)-based quantitative proteomics analysis
We commissioned Shanghai Applied Protein Technology Co., Ltd. to perform TMT-based quantitative proteomic analysis of Hep3B cells stably overexpressing *PRDM1* and the corresponding control cells, (*PRDM1* and vector cells, respectively), with three replicates per group. In brief, cells were first lysed with SDT buffer (4% (w/v) SDS, 100 mM Tris/HCl (pH 7.6), 0.1 M DTT) and trypsinized with the filter-aided proteome preparation (FASP) technique. Then, 100 μg peptide mixture of each sample was labeled using TMT reagent according to the manufacturer's instructions (Thermo Scientific, USA). A Pierce High pH Reversed-Phase Fractionation Kit (Thermo Fisher Scientific, USA) was used to fractionate samples of the TMT-labeled digests into 10 fractions via step gradient elution with increasing concentrations of acetonitrile according to the instructions. The collected fractions were desalted on C18 Cartridges (Empore™ SPE Cartridges C18 (standard density), bed I.D. 7 mm, volume 3 ml, Sigma) and concentrated by vacuum centrifugation. Each fraction was injected for LC-MS/MS analysis. LC-MS/MS analysis was performed on a Q-Exactive mass spectrometer (Thermo Scientific) that was coupled to Easy nLC (Proxeon Biosystems, now Thermo Fisher Scientific) for 60 min. The MS/MS spectra data were searched using MASCOT engine embedded into Proteome Discoverer 1.4 software. Student's *t* test was performed to identify significant differences between the *PRDM1*-overexpressing and control groups. The upregulation threshold was set at the ratio of comparison groups >1.2 and *P* value <0.05, and the downregulation at the ratio of comparison groups <0.83 and *P* value <0.05.

## Luciferase reporter assay
Luciferase reporter assays were performed according to the manufacturer's instructions. In brief, HCC cells were seeded in 24-well plates and incubated until they reached 70% confluence. They were then transfected with 1 μg of truncated or mutated *PD-L1/USP22* promoter luciferase reporter in the indicated cells along with 0.025 μg pRL-TK for normalization. Forty-eight hours after transfection, the cells were harvested in lysis buffer. Luciferase activity was measured using the Dual-Luciferase Reporter Assay System (Promega, Madison, WI) according to the manufacturer's instructions.

## ChIP assays
Hep3B and Huh7 cells transfected with *SPI1*/*PRDM1* and sh*SPI1*/sg*PRDM1* lentiviral vectors, respectively, were cross-linked with 1% formaldehyde

for 10 min at room temperature; the reaction was stopped by the addition of glycine to a final concentration of 0.125 M for another 10 min. Then, the cells were washed twice in cold PBS and harvested in lysis buffer (P2078-11, Beyotime). The samples were sonicated 20 times (30 s on/60 s off, 260 W) at 4 °C using a Diagenode Bioruptor. Samples were precleared with Protein A/G Agarose (P2078-1, Beyotime) for 30 min at 4 °C. After the 1% input sample was extracted, the samples were divided equally and incubated with an anti-SPI1 antibody (ab227835, Abcam, 5 μg/25 μg of chromatin for CHIP)/anti-BLIMP1 antibody (ab13700, Abcam, 5 μg/25 μg of chromatin for CHIP) or IgG (BS-0295P, Bioss Antibodies) conjugated to Protein A/G Agarose (P2078-1, Beyotime) at 4 °C overnight. Then, the immune complexes were washed with Low-Salt Immune Complex Wash Buffer (P2078-4, Beyotime), High-Salt Immune Complex Wash Buffer (P2078-5, Beyotime), LiCl Immune Complex Wash Buffer (P2078-6, Beyotime) in turn for 5 min at 4 °C rotation and then washed twice with TE Buffer (P2078-7, Beyotime). DNA-protein complexes were eluted with 250 mL of elution buffer (1% SDS and 0.1 M NaHCO3) and de-crosslinked by adding 0.2 M NaCl and shaking for 4 h at 65 °C. Then, the samples were digested with proteinase K, and the enriched DNA was purified by a DNA Purification Kit (D0033, Beyotime). For all ChIP experiments, qPCR analyses were performed in real time by using ABI PRISM 7900 Sequence Detection System and SYBR Green Master Mix. Threshold cycles (Ct) were determined for both immuno-precipitated DNA and DNA from the input sample. The standardized method is as follows: $\Delta Ct$ [normalized ChIP] = Ct [ChIP] − (Ct [Input] − Log2 (Input Dilution Factor)); Input Dilution Factor = (fraction of the input chromatin saved)$^{-1}$. %Input = $2^{(-\Delta Ct \text{ [normalized ChIP]})} \times 100\%$. The primers used are listed in Supplementary Table 5.

### Immunohistochemistry and immunofluorescence

For immunohistochemistry (IHC), HCC patient tissue microarrays (TMA) were produced by Zhuoli Biotech (Shanghai, China) and were stained with the indicated antibodies. Multi-color IHC assays were performed using the respective kits (Panovue, Beijing, China). Immunofluorescence images were acquired using a confocal microscope (Leica LAS AF Lite 2.6.0). Signal intensities were quantified by ImageJ 1.8.0. Antibodies for IHC and multi-color IHC were: BLIMP1 (ab198287, Abcam, 1:500), USP22 (ab195289, Abcam, 1:1000), SPI1 (ab227835, Abcam, 1:1000), PD-L1 (ab205921, Abcam, 1:1000), CD8 alpha (ab245118, Abcam, 1:1000), and GZMB (ab255598, Abcam, 1:3000). Antibodies used for immunofluorescence were USP22 (ab235923, Abcam, 1:200), SPI1 (ab88082, Abcam, 1:200), CD8 alpha (#GB11068, Servicebio, 1:200), and PD-L1 (ab213480, Abcam, 1:200).

### Co-immunoprecipitation (co-IP) assay and mass spectrometry analysis

Co-IP assays were performed to determine the interaction between USP22 and SPI1. The complexes were precipitated using protein A/G-agarose beads, followed by western blotting. Silver-stained proteins were then excised and subjected to mass spectrometry analysis. Antibodies used were: USP22 (ab195289, Abcam, 1:40) and SPI1 (ab227835, Abcam, 1:30).

### Animal models

All the animal studies were approved by the Institutional Animal Care and Use Committee of Nanjing Medical University and conducted according to protocols approved by the Ethical Committee of Nanjing Medical university. Mice were housed in specific pathogen-free (SPF) conditions, dark/light cycles: 12-h light/12-h dark (150–300 lux), ambient temperature 20–26 °C, and humidity 40–70%, ventilated four times per hour. For the immunodeficient mouse model, *Prdm1* Hepa1-6 cells, sg*Prdm1* H22 cells, and their corresponding control cells (1 × 10⁶/100 μL PBS) were inoculated into the left flanks of 4-week-old BALB/c nude male mice. Tumor size was measured using digital calipers. For the immune-competent mouse model, *Prdm1*

Hepa1-6 cells, sg*Prdm1* H22 cells, and their corresponding control cells (2 × 10⁶/100 μL PBS) were inoculated into the left flanks of 4-week-old C57BL/6 male mice. First, the above-mentioned cells were inoculated to determine the impact of an intact immune system on the *Prdm1*-induced immune response. Afterward, mouse-PD-1 mAb (BioXcell, BE0146) and IgG isotype control (BioXcell, BE0089) were employed to determine whether *Prdm1* overexpression had a synergistic effect with PD-1 mAb therapy. *Pd-l1*-knockout HCC cells were generated to evaluate whether *Prdm1*-induced immune evasion was dependent on PD-L1 expression. Mouse-CD8α mAb (BioXcell, BP0117) was also used in tumor-bearing mice to examine whether the synergistic effect of *Prdm1* overexpression and PD-1 mAb was dependent on CD8⁺ T cells. The mAb used in immune-competent mouse models was delivered via intraperitoneal injection. Tumor sizes were calculated by the volume formula $\pi/6(L \times W \times H)$ [*L*: the length (longest dimension); *W*: width (shorter dimension, parallel to the mouse body); *H*: height (diameter of tumor perpendicular to the length and width)], which is confirmed to be the best formula represented the actual volume of a wide spectrum of tumor shapes and sizes[38]. The maximal tumor weight was not exceeded 10% weight of the chosen animal as stipulated by the Ethics Committee of Nanjing Medical University. Besides, euthanasia was implemented under some circumstances: such as the expectation of death, extremely weak physiology conditions, and tumor burden that leads to ulcer and abnormal behavior. Tumors were finally harvested for flow cytometry analysis and further pathological analysis.

### Statistical analysis

Statistical analyses were performed using GraphPad Prism 8. Means between the two groups were analyzed using the Student's *t* test (unpaired two-tailed). Pearson's correlation analysis was performed to test the correlation between two variables. Survival data are presented as Kaplan–Meier survival curves, and differences between groups were evaluated using the log-rank test. Statistical significance was set at $P < 0.05$.

### Reporting summary

Further information on research design is available in the Nature Portfolio Reporting Summary linked to this article.

## Data availability

The mass spectrometry proteomics data in this study have been deposited to the ProteomeXchange Consortium via the PRIDE partner repository under accession code PXD037621. The remaining data are available within the article, Supplementary Information, or Source data file. Source data are provided with this paper.

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

## Acknowledgements

This work was supported by the Major Program of the National Natural Science Foundation of China (31930020 to X.W.), the National Natural Science Foundation of China (82203659 to Q.L., 82272791 to X.W., and 82103440 to Jing Xu), the Natural Science Foundation of Jiangsu Province (BK20220731 to Q.L.), the China Postdoctoral Science Foundation (2022M711405 and 2022TQ0130 to Q.L.), the Foundation for Talented Scholars of The First Affiliated Hospital of Nanjing Medical University (MXJL202103 to Q.L.), the Major Program of Yili Clinical Medical Research Institute (yl2021zd04 to X.W.), Postgraduate Research & Practice Innovation Program of Jiangsu Province (SJCX22_0685 to F.Y.), and Jiangsu Key Lab of Cancer Biomarkers, Prevention and Treatment, Collaborative Innovation Center for Cancer Medicine, Nanjing Medical University (to X.W.).

## Author contributions

X.W., J.T., C.Z., Y.X., and Jing Xu conceived the project and designed the research. Q.L., L.Z., and Jiali Xu performed the in vitro experiments. Q.L., W.H., and J.D. were responsible for the in vivo experiments. Q.L., W.Y., J.D., D.H., and Jing Xu drafted the manuscript. Y.D., F.Y., R.Z., J.D., S.Z., Y.Z., and T.B. performed data analysis. X.Q. and L.P. were responsible for quality control. All authors have read and approved the final manuscript.

## Competing interests

The authors declare no competing interests.
