## [Peer Review File · Nature Communications]

PRDM1/BLIMP1 induces cancer immune evasion by modulating the USP22-SPI1-PD-L1 axis in hepatocellular carcinoma cellsREVIEWER COMMENTS

Reviewer #1, expert in HCC (Remarks to the Author):

Authors employed a variety of model systems (including analyses of specimens from HCC patients) to characterize the involvement of PRDM1 in immune evasion through tumoral expression of PD-L1. Authors presented evidence that PRDM1 transcriptionally upregulated the deubiquitinating enzyme USP22 to stabilize the transcription factor SPI1, which stimulates expression of PD-L1. They have used several mouse models to establish correlations between PRDM1, PD-L1 expression and CD8+ T cell exhaustion. The correlations were also observed in human HCC samples. In addition, they provided proof of principle that PRDM1 expression in mouse models of HCC sensitizes the cancers to PD1-antibody therapy. The ms. is interesting, but there are several concerns.

1. Two effects of PRDM1 on tumor growth and PD-L1 expression needs further clarification. In Fig. 2, authors should stably knockdown PD-L1 in Hepa1-6 cells and analyze the effects of PRDM1 expression in immunocompetent mice.
2. Fig. 3, panel E, there are 4 samples (2 +IFN and 2 – IFN) but only two loading controls are included.
3. Fig. 4: In panel C, authors show that in the presence of MG132, absence of PRDM1 makes no difference in the protein levels of SPI1. Then why input SPI1 levels in panel E are different? The same problem with panel K where shUSP22 was used in the presence and absence of MG132 – The input levels in panel M do not agree.
4. Fig. 5E, SPI1 input levels are not very comparable.
5. Fig. 6: Expression of PRDM1 in the presence of PD-1-ab did not increase active CD8+ T cells. Is the tumor-inhibition not related to active T-cell?
6. Single-cell RNA-Seq: Two patients (P1 and P2) – The cell numbers (much lower in P1) are widely different - The comparison of the infiltrating cells is problematic.

Reviewer #2, expert in PD-L1 regulation (Remarks to the Author):

The authors identified a biomarker/driver of PD-L1 expression, PRDM1, and its mechanism in regulating immune function. However, it lacks clinical applicability and novelty as a therapeutic strategy, as the authors mention. It is hard to expect the wide use of this biomarker clinically nor the use of the protein/transcription factor as a therapeutic target, hindering the significance and applicability of the paper. Furthermore, it is still unclear whether PRDM1-mediated tumor immune escape depends on PD-L1 although many data were suggested.

The overall novelty of this manuscript may not reach Nat Comm's scope.

In figure 2, it is still not clear why no significant differences were observed between the PRDM1 or sgPRDM1 groups and their corresponding control groups in terms of tumor size and OS. Although the population of CD8+GZMB+ CD* T cells in the TIL was changed, the tumor growth between the PRDM1 or sgPRDM1 groups and their corresponding control groups. If the PRDM1 derived proliferation of tumor cells contributes no differences between the two groups, The number of increased tumor cells by PRDM1 would be similar to that of cytotoxic T cells decreased tumor cells. Therefore, the numbers of proliferating/apoptotic tumor cells (i.e., Ki-67, active caspase 3) should be quantified in Figure 2.

For the experiment described in Fig 2A-I, the authors can establish PD-L1-/- Hepa1-6 and H22 cell lines and perform the same in vivo experiment. They should expect to get similar results to Fig 2 J-O, although the tumor growth of shPD-L1 Hepa1-6 tumor was presented in Figure S3. This will convince readers that PRDM1-induced tumor growth is caused by PD-L1 stabilization but not other immune modulation.

In Figure 1G, Both T cells and HCC cells can secrete TNF alpha and the experiment can't distinguish

the source of TNF alpha. Thus, only T cell-produced TNF alpha should be quantified in a different experimental setting.

In Figure 4, the authors showed that PRDM1 restrains SPI1 polyubiquitination and proteasomal degradation. And then, they tried to identify a potential DUB accounting for SPI1 protein stability. In addition to DUBs, E3 ligases are also responsible for protein degradation. However, there is no rationale why E3 ligases were excluded in the process.

Reviewer #3, expert in sc-RNAseq and TiME (Remarks to the Author):

In this manuscript, Li et al found that tumoral PRDM1 overexpression is a double-edge sword in regulating tumor growth. PRDM1 overexpression inhibits cell-intrinsic cell growth in immune-deficient mouse, while promotes tumor cell immune evasion by up-regulating PD-L1 and dampening CD8+ T cell anti-tumor immune response in immune-competent mice. This interesting finding highlights the importance to study cancer biology in immune-competent context. Mechanistically, the authors found that PRDM1 enhances USP22 transcription, thus reducing SPI1 protein degradation through de-ubiquitination, which enhanced PD-L1 transcription. Multi-color immunohistochemistry revealed that PRDM1, USP22, SPI1, and PD-L1 levels were negatively correlated with CD8+ T 38 cells and the activity (GZMB+) of infiltrated CD8+ T cells. They also showed that PD-1 mAb treatment reinforced the efficacy of PRDM1-overexpressing 36 HCC immune-competent mouse models. Collectively, the authors conclude that PRDM1-USP22-SPI1 axis regulates PD-L1 levels, resulting in infiltrated CD8+ T cell exhaustion, and propose that PRDM1 overexpression combined with PD-(L)1 mAb treatment is likely to be a novel therapeutic strategy for HCC treatment.

PD-L1 is a well-established and important player in mediating tumor immune evasion by dampening anti-tumor immune response of CD8 T cells through binding to PD1. Therefore, to study the regulation of PD-L1 expression is an important and hot topic in the field. The manuscript offered a new mechanism in which PD-L1 can be regulated in HCC. Importantly, the mechanistic experiments are comprehensive and solid. Given the relatively refractory nature of HCC to immunotherapy, this finding suggests that combining PD-(L)1 blockade immunotherapy with PDRM1 overexpression is a novel strategy for HCC treatment. Overall, I recommend this manuscript to be published, but not in its current form. Many issues must be addressed before its publication.

Major issues:

1. In the last section "Single-cell analysis of intra-tumoral immune cell populations confirmed that PRDM1 overexpression potentiates T-cell exhaustion", only 2 patients were analyzed. Given the significant heterogeneity among HCC patients, the authors must increase the number of patients analyzed. This can be done with FACS analysis, not necessarily scRNA-seq.

Minor issues:

1. Line 425 typo.
2. The authors can try to address if their finding in HCC also applies to other cancer?
3. In Figure 3E, why add IFN- γ to Hep3B and Huh7 cells?
4. Figure 1K, %GZMB+ of CD8+ T cells typo
5. Figure 2P, the y-coordinate should start at 0
6. In Figure 3B, proteomics data need to be labeled better.
7. The ubiquitin-proteasome system, such as in Figure 5, SPI1 ubiquitination was determined by HA-ubiquitin immunoprecipitation and WB using an SPI1 antibody maybe more

Reviewer #4, expert in PRDM1/Blimp (Remarks to the Author):

In this paper Li et al. Investigate the relationship between the transcription factor BLIMP1 encoded by PRDM1 and the immune checkpoint inhibitor PD-1L in hepatocellular carcinoma. The authors start by showing that PD-1L is induced upon BLIMP1 overexpression in hepatocellular carcinoma (HCC) cell lines, both by western blotting and flow cytometry, and also show that this effect

is synergized by gamma interferon treatment. The authors also show the converse effect by mutating PRDM1 via CRISPR-Cas9. They also show an apparent correlation between PD-1L and PRDM1 expression in HCC using a data repository. However, the nature of the data or the depository is never described in the paper. The authors furthermore show through several experimental approaches that BLIMP1 expression has a neutral effect on the growth of HCC in animal models, since and reveal the opposing effects of BLIMP1 on proliferation and tumour immune evasion, where it appears that the anti-proliferative effects of BLIMP1 (that are well documented in the literature but not well cited in the current work) is counteracted by the induction of PD-1L and the downstream T-cell exhaustion. The authors then move on to chase the molecular pathway leading to BLIMP1 mediated induction of PD-1L and come to the conclusion that BLIMP1 induces the expression of the USP22 deubiquinating enzyme that in turn increases the stability of the transcription factor SP1 that goes on to induce the expression of PD-1L at the level of transcript. They then use single cell transcriptomics to characterize the expression levels of the mRNAs encoding the above factors in HCC and come to the conclusion that they form a cascade in the tumors. Furthermore, they perform transplantation experiment of BLIMP1 over-expressing HCC cell lines or the empty vector control, showing that whereas BLIMP1 has a neutral effect on tumour growth its overexpression leads to dramatically reduced tumour growth upon anti PD-1L immunotherapy.

My over-all impression of the work is that whereas its findings might be relevant and interesting, the authors fail to put it into the context of other work in the field of tumour immune escape vs. anti-proliferative effects of BLIMP1 on the one hand and the very well documented role of BLIMP1 as an interferon regulated gene. Given the experimental set up a much better context of the previous work published on BLIMP1 would be in order.

Additionally, the logic of the paper was extremely hard to follow. This might in part be because of a language problem (e.g. in line 456 the word „conquers“ is used instead of what would seem to mean „confers“, there are many more issues like this in the paper), however although this slowed down the reading of the paper, it was more the incompleteness of data presentation and lack of methodological description that impeded my assessment of the work.

Below are some specific points.

1. The authors refer to the factor as PRDM1 both when discussing the gene as well as the protein. While most of the PRDM family factors have the same name for both the protein as well as the gene, the human protein product of the PRDM1 gene is referred to in the literature as either PRDI-BF1 (Human positive regulatory domain I binding factor I) or BLIMP1 (B-lymphocyte induced maturation protein -1). It would be preferable for the authors to use either naming convention, as it would provide increased transparency as well as keeping with the convention in the field.
2. The chromatin immunoprecipitation results are very unconvincing. A successful ChIP-qPCR experiment for a transcription factor would normally show an enrichment of at least 20 fold over a negative control region or more. In the paper no attempt is made at looking at a negative control region and it appears that enrichment is solely reported over an IgG control. As IgG really only provides a baseline control to make sure there is no DNA contamination in the assay, this is not very informative. One or two promoter regions of expressed genes not expected to be bound by the factors assayed (SPI1 and BLIMP1) should be assayed at a minimum. Typically such genes will often show 5-10 fold enrichment over IgG. Also, reporting ChIP assays with fold enrichment rather than %-input often masks artefacts in the assay. Additionally, there is no methodological section on chromatin immunoprecipitation, and the assay can therefore not be evaluated thoroughly as it is not clear whether an antibody to BLIMP1 or SPI1 was used etc.
3. In figure 3, panel B, the authors claim to have performed a proteomic analysis but provide no explanation of what this entails. There is no description in the results, legend or the methods of any proteomic analysis. Furthermore, the figure panel is completely uninformative as it doesn't list any conditions or any proteins.
4. 3E is hard to understand. Why is there a panel for both IFN γ positive and negative PD-1L? Were these separate samples? Were they loaded equally? Are these bands from different blots?
5. There is published data showing that USP22 can directly affect the degradation of PD-1L. Why isn't this addressed by the authors?[1]

Finally, the conclusion that BLIMP1 overexpression sensitizes the cells to anti PD-1 therapy I think is an over-interpretation of the results presented (Figure 6). Rather, it looks like it is the anti PD-1 therapy that uncovers the anti-proliferative effect of BLIMP1 in these cancer cell line derived tumours. Furthermore, the tumours seem to already be sensitive to anti PD-1 therapy as judged by the survival curves presented.

I therefore regret to say that I can not recommend the publication of this manuscript in Nature Communications.

1. Wang Y, Sun Q, Mu N, et al (2020) The deubiquitinase USP22 regulates PD-L1 degradation in human cancer cells. *Cell Commun Signal* 18:112. <https://doi.org/10.1186/s12964-020-00612-y>

We sincerely appreciate all valuable comments and suggestions raised by reviewers. Now we have revised and improved our study according to the comments. All concerns have been fully addressed and the point-by-point responses are as follow:

Response to Reviewer #1, expert in HCC:

Comment 1

Two effects of PRDM1 on tumor growth and PD-L1 expression needs further clarification. In Fig. 2, authors should stably knockdown PD-L1 in Hepa1-6 cells and analyze the effects of PRDM1 expression in immunocompetent mice.

Response: Thank you for your kind advice. To confirm the functional association between BLIMP1/PRDM1-induced PD-L1 upregulation and *in vivo* tumor enlargement, we have established PD-L1^{-/-} Hepa1-6 and H22 cell lines and perform the same *in vivo* experiment in Fig. 2a and 2d. The established PD-L1^{-/-} Hepa1-6 and H22 cell lines with BLIMP1/PRDM1 overexpression vector and knockout vector, respectively, were inoculated into immune-competent mice (C57BL/6 mice). The results revealed that BLIMP1/PRDM1 overexpression in immune-competent mice inhibited tumor proliferation, and that BLIMP1/PRDM1 knockout contributed to tumor proliferation (Supplementary Fig. 3i-n). Thus, above results confirmed that BLIMP1/PRDM1 may promote tumor immune evasion by driving PD-L1 upregulation and neutralizing the anti-tumor efficacy of BLIMP1/PRDM1 overexpression.

Comment 2

Fig. 3, panel E, there are 4 samples (2 +IFN and 2 – IFN) but only two loading controls are included.

Response: Thank you very much for your helpful comments. As you said, it is confusing to set a panel for PD-L1 expression at protein levels following incubation with or without IFN- γ . We have rearranged our panels in Fig. 3e and included the other two loading controls in our revised Fig. 3e.

Comment 3

Fig. 4: In panel C, authors show that in the presence of MG132, absence of PRDM1 makes no difference in the protein levels of SPI1. Then why input SPI1 levels in panel E are different? The same problem with panel K where shUSP22 was used in the presence and absence of MG132 – The input levels in panel M do not agree.

Response: We are really grateful for your kind and insightful comments about our work. We have analyzed all the experimental procedures and found that we might neglect to add MG132 in our input samples. Hence, we have repeated experiments in the presence of MG132 and confirmed that SPI1 input levels are comparable in input samples regardless of BLIMP1 (PRDM1)/USP22 expression. Thank you very much again for helping us correct this mistake of our manuscript.

Comment 4

Fig. 5E, SPI1 input levels are not very comparable.

Response: Accordingly, we have repeated experiments in this panel in the presence of MG132 and confirmed that SPI1 input levels are comparable in input samples regardless of diverse treatment. We sincerely appreciate the reviewer's time and efforts in reviewing and improving our manuscript by raising insightful comments and suggestions.

Comment 5

Fig. 6: Expression of PRDM1 in the presence of PD-1-ab did not increase active CD8+ T cells. Is the tumor-inhibition not related to active T-cell?

Response: Thank you for your suggestion very much. Compared with IgG group in Hepa1-6 cells, PD-1 mAb treatment dramatically inhibited tumor proliferation and extended survival time regardless of BLIMP1/PRDM1 expression levels. Moreover, co-treatment with BLIMP1/PRDM1 overexpression and PD-1 mAb further impaired tumor proliferation compared with that in BLIMP1/PRDM1 overexpression or PD-1 mAb treatment alone (Fig. 6a, c, e, f and i). Hence, compared with vector group, BLIMP1/PRDM1 overexpression in the presence of PD-1 mAb contributed tumor-inhibition through impaired tumor cell proliferation and increased active CD8⁺ T cells.

However, compared with vector group in the presence of PD-1 mAb, BLIMP1/PRDM1 overexpression in the presence of PD-1 mAb mainly contributed tumor-inhibition through impaired tumor cell proliferation.

Comment 6

Single-cell RNA-Seq: Two patients (P1 and P2) – The cell numbers (much lower in P1) are widely different - The comparison of the infiltrating cells is problematic.

Response: Thank you for this valuable comment. During the process of biopsy, the patients with HCC are prone to bleeding, which makes the biopsy more difficult and the cell numbers between the two patients relatively different. Hence, we analyzed the abundance of each subpopulation between patients 1 and 2 by comparing the ratio. Meanwhile, to further validate the results of single-cell RNA-Seq, we have increased the number of HCC biopsies before PD-1 mAb-based therapies and performed multi-color immunohistochemistry (IHC). As expect, BLIMP1/PRDM1 levels were negatively correlated with CD8⁺ T cells and the activity (GZMB⁺) of infiltrated CD8⁺ T cells (Supplementary Fig. 8a, b). Meanwhile, patients with high tumoral BLIMP1/PRDM1 expression showed significant tumor shrinkage after treatment, whereas patients with low tumoral BLIMP1/PRDM1 expression showed tumor progression after treatment (Supplementary Fig. 8c, d). The results confirmed our hypothesis that BLIMP1/PRDM1 overexpression contributes to the therapeutic effects of PD-1 mAb therapy.

Response to Reviewer #2, expert in PD-L1 regulation:

Comment 1

In figure 2, it is still not clear why no significant differences were observed between the PRDM1 or sgPRDM1 groups and their corresponding control groups in terms of tumor size and OS.

Although the population of CD8⁺GZMB⁺ CD⁺ T cells in the TIL was changed, the tumor growth between the PRDM1 or sgPRDM1 groups and their corresponding control groups. If the PRDM1 derived proliferation of tumor cells contributes no

differences between the two groups,

The number of increased tumor cells by PRDM1 would be similar to that of cytotoxic T cells decreased tumor cells. Therefore, the numbers of proliferating/apoptotic tumor cells (i.e., Ki-67, active caspase 3) should be quantified in Figure 2.

Response: We are really grateful for your kind and insightful comments about our work. In our study, we found that tumoral BLIMP1/PRDM1 overexpression upregulated PD-L1 levels, dampening anti-tumor immunity in vivo, and neutralized the anti-tumor efficacy of BLIMP1/PRDM1 overexpression in immune-competent mouse models. Tumoral BLIMP1/PRDM1 overexpression is a double-edge sword in regulating tumor growth. Hence, the BLIMP1/PRDM1 and sgBLIMP1/sgPRDM1 groups and their corresponding controls showed comparable tumor sizes and OS.

Moreover, according to your suggestion, we have performed immunohistochemical analysis of Ki67 expression. No obvious differences were observed between the BLIMP1 or sgBLIMP1 groups and their corresponding controls in terms of Ki67 staining in immunocompetent mice. Nevertheless, BLIMP1/PRDM1 overexpression reduced cell proliferation, while BLIMP1/PRDM1 knockout promoted cell proliferation in immunodeficient mice (Supplementary Fig. 2b, c). These data demonstrated the dual character of BLIMP1/PRDM1, which decreases HCC cell proliferation to suppress HCC growth, while promoting HCC immune escape via regulation of PD-L1.

Comment 2

For the experiment described in Fig 2A-I, the authors can establish PD-L1/-Hepa1-6 and H22 cell lines and perform the same in vivo experiment. They should expect to get similar results to Fig 2 J-O, although the tumor growth of shPD-L1 Hepa1-6 tumor was presented in Figure S3.

This will convince readers that PRDM1-induced tumor growth is caused by PD-L1 stabilization but not other immune modulation.

Response: Thank you very much for your helpful comments. To confirm that BLIMP1/

PRDM1-induced tumor growth is caused by PD-L1 upregulation but not other immune modulation, we have established PD-L1^{-/-} Hepa1-6 and H22 cell lines. The established PD-L1^{-/-} Hepa1-6 and H22 cell lines with BLIMP1/PRDM1 overexpression vector and knockout vector, respectively, were inoculated into immune-competent mice (C57BL/6 mice). The results revealed that BLIMP1/PRDM1 overexpression in immune-competent mice inhibited tumor proliferation, and that BLIMP1/PRDM1 knockout contributed to tumor proliferation (Supplementary Fig. 3i-n). Thus, above results confirmed that BLIMP1/PRDM1 may promote tumor immune evasion by driving PD-L1 upregulation and neutralizing the anti-tumor efficacy of BLIMP1/PRDM1 overexpression.

Comment 3

In Figure 1G, Both T cells and HCC cells can secrete TNF alpha and the experiment can't distinguish the source of TNF alpha. Thus, only T cell-produced TNF alpha should be quantified in a different experimental setting.

Response: We are really grateful for your kind and insightful comments about our work. According to your suggestion, we have performed flow-cytometric analysis of CD8⁺TNFα⁺ cell content in a 3D culture system. The results revealed that BLIMP1/PRDM1 upregulation impaired the T cell-mediated tumor cell killing activity (CD8⁺TNFα⁺ cells) in the co-culture, whereas BLIMP1/PRDM1 knockout had contrasting effects (Fig. 1g).

Comment 4

In Figure 4, the authors showed that PRDM1 restrains SPI1 polyubiquitination and proteasomal degradation. And then, they tried to identify a potential DUB accounting for SPI1 protein stability. In addition to DUBs, E3 ligases are also responsible for protein degradation. However, there is no rationale why E3 ligases were excluded in the process.

Response: Thank you very much for your comments to the details of our article. Our results in Fig. 3a-e indicated that BLIMP1/PRDM1 contributed to SPI1

deubiquitination and restrained its proteasomal degradation. Hence, we wondered whether there is a potential deubiquitinating enzyme (DUB) accounting for SPI1 deubiquitination in BLIMP1/PRDM1-overexpressing cells. Meanwhile, DUB siRNA library, which is designed for effective and convenient siRNA to knock down the entire DUB family, can be used to screen this potential DUB efficiently. Therefore, we employed DUB siRNA library to try to identify the DUB accounting for SPI1 protein stability. Meanwhile, BLIMP1/PRDM1 overexpression increased the expression levels of USP22, whereas BLIMP1/PRDM1 knockout decreased the expression levels of USP22, which is consistent with our proteomic analysis results. Thus, BLIMP1/PRDM1 was inferred to stabilize SPI1 via USP22. As you said, E3 ligases are also responsible for protein degradation. We could not fully exclude the possibility that BLIMP1/PRDM1 may also influence SPI1 protein stability through a specific E3 ligases, but, at least, we confirmed that USP22 is required for SPI1 deubiquitination in HCC. Hence, the underlying mechanisms need to be investigated further. This key point has also been clearly discussed in our revised manuscript.

Response to Reviewer #3, expert in sc-RNAseq and TIME:

Comment 1

In the last section “Single-cell analysis of intra-tumoral immune cell populations confirmed that PRDM1 overexpression potentiates T-cell exhaustion”, only 2 patients were analyzed. Given the significant heterogeneity among HCC patients, the authors must increase the number of patients analyzed. This can be done with FACS analysis, not necessarily scRNA-seq.

Response: Thanks to the reviewer’s constructive comment. Per the reviewer’s suggestion, we have added the number of HCC biopsies before PD-1 mAb-based therapies. Because these specimens were fixed with formaldehyde, we thus performed multi-color immunohistochemistry (IHC) to analyze the activity (GZMB⁺) of infiltrated CD8⁺ T cells instead of FACS analysis. As expect, BLIMP1/PRDM1 levels were negatively correlated with CD8⁺ T cells and the activity (GZMB⁺) of infiltrated CD8⁺ T cells (Supplementary Fig. 8a, b). Meanwhile, patients with high tumoral

BLIMP1/PRDM1 expression showed significant tumor shrinkage after treatment, whereas patients with low tumoral BLIMP1/PRDM1 expression showed tumor progression after treatment (Supplementary Fig. 8c, d). The results confirmed our hypothesis that BLIMP1/PRDM1 overexpression contributes to the therapeutic effects of PD-1 mAb therapy.

Comment 2

Line 425 typo.

Response: We have to say sorry about the troubles brought by our typos and thank you for your keen comments. We have corrected it in our revised manuscript.

Comment 3

The authors can try to address if their finding in HCC also applies to other cancer?

Response: We are really grateful for your kind and insightful comments about our work. We fully agree with the reviewer that it would be of great interest if our finding in HCC also applies to other cancers. In the future, we will focus on this item and make more in-depth investigation.

Comment 4

In Figure 3E, why add IFN- γ to Hep3B and Huh7 cells?

Response: Thank you very much for your helpful comment. Recent studies revealed that silenced tumoral PD-L1 expression in many malignancies could be significantly induced under IFN- γ stimulation. The reason for adding IFN- γ to Hep3B and Huh7 cells is that we wanted to study whether BLIMP1/PRDM1 regulates both constitutive and induced (in response to IFN- γ) PD-L1 expression.

Comment 5

Figure 1K, %GZMB+ of CD8+ T cells typo.

Response: We have to say sorry about the troubles brought by our typos and thank you for your keen comments. We have corrected it in our revised Fig. 1k.

Comment 6

Figure 2P, the y-coordinate should start at 0.

Response: Thank you very much for your comments to the details of our article. We have revised it in our revised Fig. 2p.

Comment 7

In Figure 3B, proteomics data need to be labeled better.

Response: Thank you for your kind advice. To make the proteomics data labeled better, we have simplified our heatmap, and displayed the top 20 upregulated proteins and the top 20 downregulated proteins between BLIMP1/PRDM1-overexpressing Hep3B cells and control cells. Meanwhile, we have attached our proteomics data as Supplementary Table 1 to display the differential proteins.

Comment 8

The ubiquitin-proteasome system, such as in Figure 5, SPI1 ubiquitination was determined by HA-ubiquitin immunoprecipitation and WB using an SPI1 antibody maybe more.

Response: We sincerely appreciate the reviewer's time and efforts in reviewing and improving our manuscript by raising insightful comments and suggestions. However, this sentence seems to be incomplete. We don't understand the meaning of this sentence.

Response to Reviewer #4, expert in PRDM1/Blimp:

Comment 1

The authors refer to the factor as PRDM1 both when discussing the gene as well as the protein. While most of the PRDM family factors have the same name for both the protein as well as the gene, the human protein product of the PRDM1 gene is referred to in the literature as either PRDI-BF1 (Human positive regulatory domain I binding factor I) or BLIMP1 (B-lymphocyte induced maturation protein -1). It would be preferable for the authors to use either

naming convention, as it would provide increased transparency as well as keeping with the convention in the field.

Response: Thank you very much for your comments to the details of our article. According to your suggestion, we have used BLIMP1 (the human protein product of the PRDM1 gene) instead of PRDM1 in our revised manuscript.

Comment 2

The chromatin immunoprecipitation results are very unconvincing. A successful ChIP-qPCR experiment for a transcription factor would normally show an enrichment of at least 20 fold over a negative control region or more. In the paper no attempt is made at looking at a negative control region and it appears that enrichment is solely reported over an IgG control. As IgG really only provides a baseline control to make sure there is no DNA contamination in the assay, this is not very informative. One or two promoter regions of expressed genes not expected to be bound by the factors assayed (SPI1 and BLIMP1) should be assayed at a minimum. Typically such genes will often show 5-10 fold enrichment over IgG. Also, reporting ChIP assays with fold enrichment rather than %-input often masks artefacts in the assay. Additionally, there is no methodological section on chromatin immunoprecipitation, and the assay can therefore not be evaluated thoroughly as it is not clear whether an antibody to BLIMP1 or SPI1 was used etc.

Response: We thank the reviewer for this constructive comment. We realized that it is really confusing to equal control or shNC group to 1 regardless of an antibody to BLIMP1/SPI1 or IgG was used. Per the reviewer's suggestion, we have reported ChIP experiment with enrichment (% of input) according to your advice. α -satellite, which is not expected to be bound by the factors assayed (SPI1 and BLIMP1), has also been employed as a negative control. As you said, the results of ChIP-qPCR experiment for transcription factor (SPI1 and BLIMP1) actually showed an enrichment of at least 20-fold over a negative control (Fig. 3i, j). Additionally, we felt sorry that methodological section on chromatin immunoprecipitation was missed. We have described this part in our revised manuscript.

Comment 3

In figure 3, panel B, the authors claim to have performed a proteomic analysis but provide no explanation of what this entails. There is no description in the results, legend or the methods of any proteomic analysis. Furthermore, the figure panel is completely uninformative as it doesn't list any conditions or any proteins.

Response: Thank you very much for the comment. Per the reviewer's suggestion, we have added the methodology of proteomics in the materials and methods section and described more details in our results and legend. To make the proteomics data labeled better, we have simplified our heatmap, and displayed the top 20 upregulated proteins and the top 20 downregulated proteins between BLIMP1/PRDM1-overexpressing Hep3B cells and control cells (Fig. 3b). Meanwhile, we have attached our proteomics data as Supplementary Table 1 to display the differential proteins.

Comment 4

3E is hard to understand. Why is there a panel for both IFNgamma positive and negative PD1-1 L? Were these separate samples? Were they loaded equally? Are these bands from different blots?

Response: Thank you very much for your helpful comments. Actually, these were separate samples and from different blots. We realized that it is confusing to set a panel for PD-L1 expression at protein levels following incubation with or without IFN- γ . We have rearranged our panels in Fig. 3e and included the other two loading controls in our revised Fig. 3e.

Comment 5

There is published data showing that USP22 can directly affect the degradation of PD-1L. Why isn't this addressed by the authors?

Response: Thank you very much for your comments to the details of our article. As Wang et.al reported, USP22 could regulate PD-L1 expression through a direct and indirect way¹. In our study, we firstly found that BLIMP1/PRDM1 mainly contributed

to PD-L1 upregulation at the transcriptional level instead of post-translational level in HCC. Subsequently, we validated that BLIMP1/PRDM1 can enhance the transcription of USP22, thus reducing SPI1 protein degradation through deubiquitination, which enhances PD-L1 transcription. Actually, considering that USP22 is mainly localised in HCC cell nuclei and PD-L1 is mainly localised in HCC cell membrane. We thought that USP22 mainly regulate PD-L1 expression through an indirect way. According to your suggestion, we have also discussed this issue in Discussion Section and cited this article in our revised manuscript. Thank you very much again for helping us improve the quality of our manuscript.

Comment 6

Finally, the conclusion that BLIMP1 overexpression sensitizes the cells to anti PD-1 therapy I think is an over-interpretation of the results presented (Figure 6). Rather, it looks like it is the anti PD-1 therapy that uncovers the anti-proliferative effect of BLIMP1 in these cancer cell line derived tumours. Furthermore, the tumours seem to already be sensitive to anti PD-1 therapy as judged by the survival curves presented.

Response: Thank you very much for the comment. As you said, PD-1 mAb treatment dramatically inhibited tumor proliferation and extended survival time compared to the IgG group in Hepa1-6 and H22 cells. More importantly, combined treatment with BLIMP1/PRDM1 overexpression and PD-1 mAb further impaired tumor proliferation and prolonged survival time compared with that in BLIMP1/PRDM1 overexpression alone. Nevertheless, co-treatment with BLIMP1/PRDM1 knockout and PD-1 mAb had a limited effect on tumor growth and survival curves compared with that in BLIMP1/PRDM1 knockout alone. From a clinical point of view, these results remind us that high BLIMP1/PRDM1-overexpressing tumors are more sensitive to anti-PD-1 treatment and BLIMP1/PRDM1 overexpression enhances the efficacy of PD-1 mAb, providing a promising therapeutic strategy for HCC treatment.

References:

1. Wang Y, Sun Q, Mu N, Sun X, Wang Y, Fan S, *et al.* The deubiquitinase USP22 regulates PD-L1 degradation in human cancer cells. *Cell communication and signaling* : *CCS* 2020, **18**(1): 112.

We acknowledge the reviewers' comments and suggestions very much, which are valuable in improving the quality of our manuscript.

REVIEWER COMMENTS

Reviewer #1 (Remarks to the Author):

Authors have addressed some of the concerns. However, several western blots lack quantifications and statistics. It is unclear how many times the experiments were done. It will be important to include the info on repeats along with statistics in the figure legends.

Reviewer #2 (Remarks to the Author):

The authors addressed all my questions and improved the quality of data and findings' reliability significantly.

Reviewer #3 (Remarks to the Author):

The manuscript offered a new mechanism in which PD-L1 can be regulated in HCC. The mechanistic experiments are comprehensive and solid. In this version, the authors have solved my major concerns. Hence, I recommend the publication of this manuscript in Nature Communications.

Reviewer #4 (Remarks to the Author):

In this paper, where Li et al. investigate the relationship between the transcription factor BLIMP1 encoded by PRDM1 and the immune checkpoint inhibitor PD-1L in hepatocellular carcinoma, it is still my impression that, although the authors present some relevant and interesting findings, my enthusiasm is dampened by the lack of clarity of data presentation as well as failure to put the finding into proper context, e.g. in the case of the role of BLIMP1 in tumour immune evasion, interferon regulation and regulation of proliferation.

The manuscript still is very hard to follow, the figure legends and methods are incomplete, and thus the experimental set up is hard to assess.

The authors have attempted to address my specific comments, but with mixed results. Please find my responses below:

1. The authors now wrongly refer to the PRDM1 gene as BLIMP1, and e.g. sgRNAs against PRDM1 sgBLIMP1, which is confusing.

2. Alas, the ChIP data is still presented in an unconvincing manner. The typical chromatin recovery of a transcription factor ChIP is less than 1% but figures 3i-j and 5p-g show recovery to be in the range of 1-50% of input. Furthermore. As stated in my comment the proper way to negatively control for immunoprecipitation artefacts in ChIP is to include one or, preferably more, promoter regions of expressed genes not expected to be bound by BLIMP1 or SPI1. Alpha satellite regions are always expected to show a low signal as they preferentially precipitate out with the insoluble fraction after sonication prior to the immunoprecipitation step.

This is absolutely critical for the interpretation of the experiments. Also, the authors have now included a rudimentary description of the ChIP methodology. However, the section looks inaccurate, as the authors state that the cells were cross-linked in 37% formaldehyde (a typical concentration is 0.4-1%), no antibody quantities were indicated and furthermore, it is hard to assess how the cells were treated/transfected etc. prior to performing ChIP. It is therefore extremely hard to assess the validity of the data or its interpretation.

3. Thank you for providing the methods section for this proteomic assessment. However, the legend and the main text still fails to clarify how the experiment was performed, what controls were used and how the samples were treated prior to mass spectrometry. The text states: „100 µg peptide mixture of each sample was labeled... „, without actually explaining what these samples are that are being referred to in the text. Thus, one needs to make a guess in order to interpret the experiment. I do not

believe this stands up to the standard of Nature Communications.

4. OK.

5. OK.

6. Thank you very much for the explanation. I think this might be stated more clearly in the paper, as the text still does not provide a clear distinction between the two presumed effects of BLIMP1, anti proliferative and the promotion of immune evasion.

Furthermore, I am not sure whether the authors mean that ectopic expression of BLIMP1 is a promising therapeutic strategy for HCC treatment? Insight into the mechanism of immune evasion can be highly relevant, but to state that the ectopic expression of transcription factors are a promising therapeutic strategy for HCC treatment seems like an exaggeration, unless clarified with how this is envisioned by the authors.

We sincerely appreciate all valuable comments and suggestions raised by reviewers. Now we have revised and improved our study according to the comments. All concerns have been fully addressed and the point-by-point responses are as follow:

Response to Reviewer #1:

Comment 1

Authors have addressed some of the concerns. However, several western blots lack quantifications and statistics. It is unclear how many times the experiments were done. It will be important to include the info on repeats along with statistics in the figure legends.

Response: Thank you for your kind advice. According to your suggestion, we have quantified the results of the western blots in our study. Meanwhile, we have also included the information on repeats along with statistics in our figure legends. Thank you very much again for helping us improve the quality of our manuscript.

Response to Reviewer #2:

Comment 1

The authors addressed all my questions and improved the quality of data and findings' reliability significantly.

Response: Our deepest gratitude goes to you for your careful work and thoughtful suggestions that have helped improve this paper substantially.

Response to Reviewer #3:

Comment 1

The manuscript offered a new mechanism in which PD-L1 can be regulated in HCC. The mechanistic experiments are comprehensive and solid. In this version, the authors have solved my major concerns. Hence, I recommend the publication of this manuscript in Nature Communications.

Response: We thank the reviewer for the time and effort committed towards improving our manuscript.

Response to Reviewer #4:

Comment 1

The authors now wrongly refer to the PRDM1 gene as BLIMP1, and e.g. sgRNAs against PRDM1 as sgBLIMP1, which is confusing.

Response: Thank you very much for your comment to the details of our article. According to your suggestion, we have referred to the gene as PRDM1 and the protein as BLIMP1 in our revised manuscript.

Comment 2

The ChIP data is still presented in an unconvincing manner. The typical chromatin recovery of a transcription factor ChIP is less than 1% but figures 3i-j and 5p-g show recovery to be in the range of 1-50% of input. Furthermore. As stated in my comment the proper way to negatively control for immunoprecipitation artefacts in ChIP is to include one or, preferably more, promoter regions of expressed genes not expected to be bound by BLIMP1 or SPI1. Alpha satellite regions are always expected to show a low signal as they preferentially precipitate out with the insoluble fraction after sonication prior to the immunoprecipitation step.

This is absolutely critical for the interpretation of the experiments. Also, the authors have now included a rudimentary description of the ChIP methodology. However, the section looks inaccurate, as the authors state that the cells were cross-linked in 37% formaldehyde (a typical concentration is 0.4-1%), no antibody quantities were indicated and furthermore, it is hard to assess how the cells were treated/transfected etc. prior to performing ChIP. It is therefore extremely hard to assess the validity of the data or its interpretation.

Response: We are really grateful for your kind and insightful comments about our CHIP assays. We realized that we have miscalculated the chromatin recovery during DNA content standardization and the chromatin recovery increased by approximately two orders of magnitude ($\sim 2^{6.67}$). The corrected standardized method is as follows: ΔCt [normalized ChIP] = Ct [ChIP] - (Ct [Input] - Log₂ (Input Dilution Factor)); Input Dilution Factor = (fraction of the input chromatin saved)⁻¹. % Input = $2^{(-\Delta Ct$ [normalized ChIP])

x 100%. Per the reviewer's suggestion, we have also included two promoter regions of PD-L1/USP22 not expected to be bound by SPI1/BLIMP1 as negative controls. Accordingly, we have repeated our CHIP assays and analyzed the data according to the corrected standardized method.

Additionally, we felt sorry for the rudimentary description of the ChIP methodology. We have described this part in our revised manuscript. According to the manufacturer's instructions, we added 550 μ L 37% formaldehyde into 20 mL DMEM medium. The final concentration of formaldehyde is approximately 1%. We have revised our previous description of the concentration of formaldehyde, which was quite misleading. Moreover, the antibody quantities, how the cells were treated/transfected prior to performing ChIP, and other details were explained in our revised manuscript. We sincerely appreciate the reviewer's time and efforts in reviewing and improving our manuscript by raising insightful comments and suggestions.

ChIP assays

Hep3B and Huh7 cells transfected with SPI1/PRDM1 and shSPI1/sgPRDM1 lentiviral vectors, respectively, were cross-linked with 1% formaldehyde for 10 min at room temperature; the reaction was stopped by the addition of glycine to a final concentration of 0.125 M for another 10 min. Then, the cells were washed twice in cold PBS and harvested in lysis buffer (P2078-11, Beyotime). The samples were sonicated 20 times (30 s on/60 s off, 260 W) at 4°C using a Diagenode Bioruptor. Samples were precleared with Protein A/G Agarose (P2078-1, Beyotime) for 30 min at 4°C. After the 1% input sample was extracted, the samples were divided equally and incubated with an anti-SPI1 antibody (ab227835, Abcam, 5 μ g/25 μ g of chromatin for CHIP)/anti-BLIMP1 antibody (ab13700, Abcam, 5 μ g/25 μ g of chromatin for CHIP) or IgG (BS-0295P, Bioss Antibodies) conjugated to Protein A/G Agarose (P2078-1, Beyotime) at 4°C overnight. Then, the immune complexes were washed with Low-Salt Immune Complex Wash Buffer (P2078-4, Beyotime), High-Salt Immune Complex Wash Buffer (P2078-5, Beyotime), LiCl Immune Complex Wash Buffer (P2078-6, Beyotime) in turn for 5 min at 4°C rotation and then washed twice with TE Buffer (P2078-7, Beyotime). DNA-

protein complexes were eluted with 250 mL of elution buffer (1% SDS and 0.1 M NaHCO₃) and de-crosslinked by adding 0.2 M NaCl and shaking for 4 h at 65°C. Then, the samples were digested with proteinase K, and the enriched DNA was purified by a DNA Purification Kit (D0033, Beyotime). For all ChIP experiments, qPCR analyses were performed in real time by using ABI PRISM 7900 Sequence Detection System and SYBR Green Master Mix. Threshold cycles (Ct) were determined for both immunoprecipitated DNA and DNA from the input sample. The standardized method is as follows: $\Delta Ct [\text{normalized ChIP}] = Ct [\text{ChIP}] - (Ct [\text{Input}] - \text{Log}_2 (\text{Input Dilution Factor}))$; $\text{Input Dilution Factor} = (\text{fraction of the input chromatin saved})^{-1}$. $\% \text{Input} = 2^{(-\Delta Ct [\text{normalized ChIP}])} \times 100\%$. The primers used are listed in Supplementary Table 5.

Comment 3

Thank you for providing the methods section for this proteomic assessment. However, the legend and the main text still fails to clarify how the experiment was performed, what controls were used and how the samples were treated prior to mass spectrometry. The text states: „100 µg peptide mixture of each sample was labeled... „, without actually explaining what these samples are that are being referred to in the text. Thus, one needs to make a guess in order to interpret the experiment.

Response: Thank you very much for the comment. Per the reviewer's suggestion, we have revised the methodology of proteomics in the materials and methods section and described more details in our legend and the main text.

Tandem mass tag (TMT)-based quantitative proteomics analysis

We commissioned Shanghai Applied Protein Technology Co., Ltd. to perform TMT-based quantitative proteomic analysis of Hep3B cells stably overexpressing PRDM1 and the corresponding control cells, (PRDM1 and vector cells, respectively), with three replicates per group. In brief, cells were first lysed with SDT buffer (4% (w/v) SDS, 100 mM Tris/HCl (pH 7.6), 0.1 M DTT) and trypsinized with the filter-aided proteome preparation (FASP) technique. Then, 100 µg peptide mixture of each sample was

labeled using TMT reagent according to the manufacturer's instructions (Thermo Scientific, USA). A Pierce High pH Reversed-Phase Fractionation Kit (Thermo Fisher Scientific, USA) was used to fractionate samples of the TMT-labeled digests into 10 fractions via step gradient elution with increasing concentrations of acetonitrile according to the instructions. The collected fractions were desalted on C18 Cartridges (Empore™ SPE Cartridges C18 (standard density), bed I.D. 7 mm, volume 3 ml, Sigma) and concentrated by vacuum centrifugation. Each fraction was injected for LC-MS/MS analysis. LC-MS/MS analysis was performed on a Q-Exactive mass spectrometer (Thermo Scientific) that was coupled to Easy nLC (Proxeon Biosystems, now Thermo Fisher Scientific) for 60 min. The MS/MS spectra data were searched using MASCOT engine embedded into Proteome Discoverer 1.4 software. Student's t-test was performed to identify significant differences between the PRDM1-overexpressing and control groups. The upregulation threshold was set at the ratio of comparison groups >1.2 and $p\text{-value} < 0.05$, and the downregulation at the ratio of comparison groups <0.83 and $p\text{-value} < 0.05$.

Comment 4

Thank you very much for the explanation. I think this might be stated more clearly in the paper, as the text still does not provide a clear distinction between the two presumed effects of BLIMP1, anti proliferative and the promotion of immune evasion. Furthermore, I am not sure whether the authors mean that ectopic expression of BLIMP1 is a promising therapeutic strategy for HCC treatment? Insight into the mechanism of immune evasion can be highly relevant, but to state that the ectopic expression of transcription factors are a promising therapeutic strategy for HCC treatment seems like an exaggeration, unless clarified with how this is envisioned by the authors.

Response: Thank you very much for your helpful comments. As stated in our study, we found that tumoral PRDM1/BLIMP1 overexpression is a double-edge sword in regulating tumor growth. PRDM1/BLIMP1 overexpression inhibits cell-intrinsic cell growth, while promotes tumor cell immune evasion by up-regulating PD-L1 and

dampening CD8⁺ T cell anti-tumor immune response simultaneously. This finding provides the theoretical basis for synergistic effect of PRDM1/BLIMP1 overexpression and PD-(L)1 mAb therapy. The underlying mechanism that separates the two presumed effects of BLIMP1 needs to be investigated further. Per the reviewer's suggestion, we have also discussed this issue in Discussion Section.

Our study indicated that PRDM1/BLIMP1 expression levels may be used to predict PD-(L)1 mAb therapy efficacy in HCC. Meanwhile, PRDM1/BLIMP1 overexpression combined with PD-(L)1 mAb treatment provides a promising therapeutic strategy for the treatment of patients with HCC. Ideal therapeutic vectors carrying PRDM1 gene are currently needed to be developed to evaluate their safety and potential to serve as vital anti-tumor drugs, for synergistically increasing the efficacy of PD-(L)1-based therapies. Nowadays, adeno-associated virus (AAV) vectors are the leading platform for gene delivery for the treatment of a variety of human diseases, including cancer. Preclinical and clinical successes in AAV-mediated gene addition, gene replacement, gene silencing and gene editing in specific tissues or cells have helped AAV gain popularity as the ideal therapeutic vector. As of 26 July 2022, there were 292 interventional clinical trials involving AAV registered at [ClinicalTrials.gov](https://clinicaltrials.gov). Some vectorized AAV serotypes have gained regulatory approval for commercial use in patients, such as AAV1 (Glybera; uniQure) and AAV2 (Luxturna; Spark Therapeutics). More efficient capsids are increasingly being utilized in trials, such as AAV8 and AAV9, which are liver-specific¹. Although the clinical success of AAV gene therapy is encouraging, we must acknowledge the challenges of this gene delivery platform, such as the large-scale vector manufacturing and cost, the vector quality control and assay standardization, and the immunological barriers to AAV gene delivery¹⁻³. Fortunately, these challenges are being addressed by a growing field that encompasses multidisciplinary expertise. Our hope is that as the AAV field continues to expand, a multidisciplinary approach to gene therapy drug development will continue to be fostered. We also expect that our study may provide a promising therapeutic strategy for HCC treatment through AAV gene delivery.

References:

- 1 Wang, D., Tai, P. W. L. & Gao, G. Adeno-associated virus vector as a platform for gene therapy delivery. *Nature reviews. Drug discovery* **18**, 358-378, doi:10.1038/s41573-019-0012-9 (2019).
- 2 Lugin, M. L., Lee, R. T. & Kwon, Y. J. Synthetically Engineered Adeno-Associated Virus for Efficient, Safe, and Versatile Gene Therapy Applications. *ACS Nano* **14**, 14262-14283, doi:10.1021/acsnano.0c03850 (2020).
- 3 Nidetz, N. F. *et al.* Adeno-associated viral vector-mediated immune responses: Understanding barriers to gene delivery. *Pharmacol Ther* **207**, 107453, doi:10.1016/j.pharmthera.2019.107453 (2020).

We acknowledge the reviewers' comments and suggestions very much, which are valuable in improving the quality of our manuscript.

REVIEWERS' COMMENTS

Reviewer #4 (Remarks to the Author):

The authors have now addressed all my concerns with the data reporting, showing that their CHIP data supports their conclusions, significantly improving the reliability of their findings. They have now also included thorough description of the experimental procedures the manuscript. I therefore have no further comments/concerns.

Reviewer #4 (Remarks to the Author):

The authors have now addressed all my concerns with the data reporting, showing that their ChIP data supports their conclusions, significantly improving the reliability of their findings. They have now also included thorough description of the experimental procedures the manuscript. I therefore have no further comments/concerns.

Response: Thanks for your positive and constructive comments. We appreciate the time and effort you have dedicated to improving our paper.